# LIGHT SCHRÖDINGER BRIDGE

**Alexander Korotin**[*,1,2], **Nikita Gushchin**[*,1], **Evgeny Burnaev**[1,2].
[1]Skolkovo Institute of Science and Technology,   [2]Artificial Intelligence Research Institute
a.korotin@skoltech.ru, n.gushchin@skoltech.ru

## ABSTRACT

Despite the recent advances in the field of computational Schrödinger Bridges (SB), most existing SB solvers are still heavy-weighted and require complex optimization of several neural networks. It turns out that there is no principal solver which plays the role of simple-yet-effective baseline for SB just like, e.g., $k$-means method in clustering, logistic regression in classification or Sinkhorn algorithm in discrete optimal transport. We address this issue and propose a novel fast and simple SB solver. Our development is a smart combination of two ideas which recently appeared in the field: (a) parameterization of the Schrödinger potentials with sum-exp quadratic functions and (b) viewing the log-Schrödinger potentials as the energy functions. We show that combined together these ideas yield a lightweight, simulation-free and theoretically justified SB solver with a simple straightforward optimization objective. As a result, it allows solving SB in moderate dimensions in a matter of minutes on CPU without a painful hyperparameter selection. Our light solver resembles the Gaussian mixture model which is widely used for density estimation. Inspired by this similarity, we also prove an important theoretical result showing that our light solver is a universal approximator of SBs. Furthemore, we conduct the analysis of the generalization error of our light solver. The code for our solver can be found at https://github.com/ngushchin/LightSB.

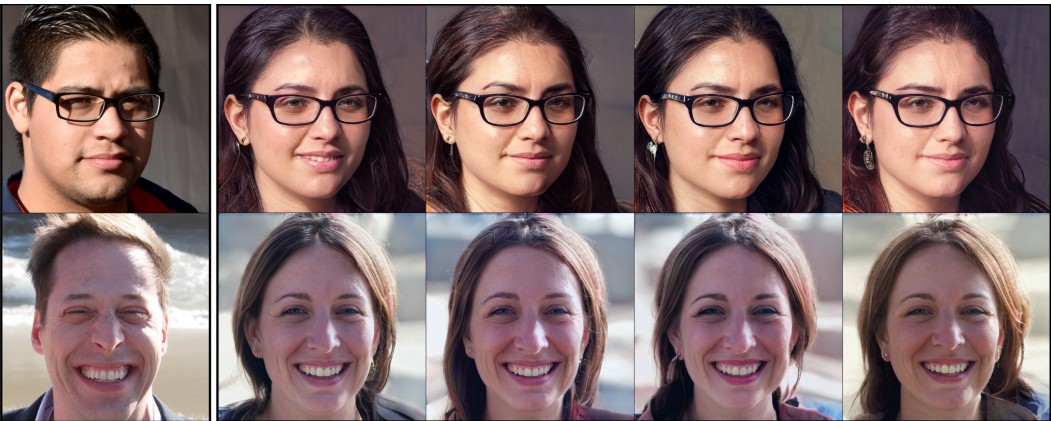

Figure 1: Unpaired *male → female* translation by our LightSB solver applied in the latent space of ALAE for 1024x1024 FFHQ images. *Our LightSB converges on 4 cpu cores in less than 1 minute.*

## 1 INTRODUCTION

Over the last several years, there has been a considerable progress in developing the computational approaches for solving the Schrödinger Bridge problem (Schrödinger, 1931; 1932, SB), which is also known as the dynamic version of the Entropic Optimal Transport (Cuturi, 2013, EOT) problem. The SB problem requires finding the diffusion process between two given distributions that is maximally similar to some given prior process. In turn, EOT problem is a simplification in which a user is interested only in the joint marginal distribution on the first and the last steps of the diffusion.

---

[*]Equal contribution.

Historically, the majority of studies in EOT/SB sub-field of machine learning are concentrated around studying EOT between **discrete** probability distributions, see (Peyré et al., 2019) for a survey. Inspired by the recent developments in the field of generative modeling via diffusions (Ho et al., 2020; Rombach et al., 2022) and the diffusion-related nature of SB, various researches started developing solvers for a **continuous** setups of EOT/SB, see (Gushchin et al., 2023b) for a survey. This implies that a learner has only a sample access to the (continuous) distributions and based on it has to recover the entire SB process (e.g., its drift) between the entire distributions. Such solvers are straightforwardly applicable to image generation (Wang et al., 2021; De Bortoli et al., 2021) and image-to-image translation tasks (Gushchin et al., 2023a) as well as to important smaller scale data transfer tasks, e.g., with the biological data (Vargas et al., 2021; Koshizuka & Sato, 2022).

Almost all existing EOT/SB solvers (see §4) have complex neural networks parameterization and many hyper-parameters; they expectedly require time-consuming training/inference procedures. Although this is unavoidable in large-scale generative modeling tasks, these techniques look too heavy-weighted when a user just wants to learn EOT/SB between some moderate-dimensional data distributions, e.g., those appearing in perspective biological applications of OT (Tong et al., 2020; Vargas et al., 2021; Koshizuka & Sato, 2022; Bunne et al., 2022; 2023; Tong et al., 2023). In this paper, we address this issue and present the following **main contributions**:

1. We propose a novel light solver for continuous SB with the Wiener prior, i.e., EOT with the quadratic transport cost. Our solver has a straightforward non-minimax learning objective and uses the Gaussian mixture parameterization for the EOT/SB (§3.1, 3.2). This development allows us to solve EOT/SB between distributions in moderate dimensions in a matter of minutes thanks to avoiding time-consuming max-min optimization, simulation of the full process trajectories, iterative learning and MCMC techniques which are in use in existing continuous solvers (§4).

2. We show that our novel light solver provably satisfies the universal approximation property for EOT/SB between the distributions supported on compact sets (§3.3).

3. We derive the finite sample learning guarantees for our solver. We show that the estimation error vanishes at the standard parametric rate with the increase of the sample size (Appendix A).

4. We demonstrate the performance of our light solver in a series of synthetic and real-data experiments (§5), including the ones with the real biological data (§5.3) considered in related works.

Our light solver exploits the recent advances in the field of EOT/SB, namely, using the log-sum-exp quadratic functions to parameterize Schrödinger potentials for constructing EOT/SB benchmark distributions (Gushchin et al., 2023b) and viewing EOT as the energy-based model (Mokrov et al., 2024) which optimizes a certain Kullback-Leibler divergence. We discuss this in §3.1.

## 2 BACKGROUND: SCHRÖDINGER BRIDGES

In this section, we recall the main properties of the Schrödinger Bridge (SB) problem with the Wiener prior. We begin with discussing the Entropic Optimal Transport (Cuturi, 2013; Genevay, 2019), which is known as the *static* SB formulation. Next, we recall the *dynamic* Schrödinger bridge formulation and its relation to EOT (Léonard, 2013; Chen et al., 2016). Finally, we summarize the aspects of the practical setup for learning EOT/SB which we consider throughout the paper.

We work in the $D$-dimensional Euclidean space $(\mathbb{R}^D, \|\cdot\|)$. We use $\mathcal{P}_{2,ac}(\mathbb{R}^D)$ to denote the set of absolutely continuous Borel probability distributions on $\mathbb{R}^D$ which have a finite second moment. For any $p \in \mathcal{P}_{2,ac}(\mathbb{R}^D)$, we write $p(x)$ to denote its density at a point $x \in \mathbb{R}^D$. In what follows, KL is a short notation for the well-celebrated Kullback-Leibler divergence, and $\mathcal{N}(x|r, S)$ is the density at a point $x \in \mathbb{R}^D$ of the normal distribution with mean $r \in \mathbb{R}^D$ and covariance $0 \prec S \in \mathbb{R}^{D \times D}$.

**Entropic Optimal Transport (EOT) with the quadratic cost.** Assume that $p_0 \in \mathcal{P}_{2,ac}(\mathcal{X})$, $p_1 \in \mathcal{P}_{2,ac}(\mathcal{Y})$ have finite entropy. For $\epsilon > 0$, the EOT problem between them is to find the minimizer of

$$\min_{\pi \in \Pi(p_0, p_1)} \left\{ \int_{\mathbb{R}^D} \int_{\mathbb{R}^D} \frac{1}{2} \|x_0 - x_1\|^2 \pi(x_0, x_1) dx_0 dx_1 + \epsilon \text{KL}\left(\pi \| p_0 \times p_1\right) \right\}, \quad (1)$$

where $\Pi(p_0, p_1)$ is the set of probability distributions (transport plans) on $\mathbb{R}^D \times \mathbb{R}^D$ whose marginals are $p_0$ and $p_1$, respectively, and $p_0 \times p_1$ is the product distribution. KL in (1) is assumed to be equal to $+\infty$ if $\pi$ is not absolutely continuous. Therefore, one can consider only absolutely continuous $\pi$. The minimizer $\pi^*$ of (1) exists; it is unique and called the *EOT plan*.

**(Dynamic) Schrödinger Bridge with the Wiener prior.** We employ $\Omega$ as the space of $\mathbb{R}^D$-valued functions of time $t \in [0, 1]$ describing trajectories in $\mathbb{R}^D$ which start at time $t = 0$ and end at $t = 1$. In turn, we use $\mathcal{P}(\Omega)$ to denote the set of probability distributions on $\Omega$, i.e., stochastic processes. We use $dW_t$ to denote the differential of the standard Wiener process. For a process $T \in \mathcal{P}(\Omega)$, we denote its joint distribution at $t = 0, 1$ by $\pi^T \in \mathcal{P}(\mathbb{R}^D \times \mathbb{R}^D)$. In turn, we use $T_{|x_0,x_1}$ to denote the distribution of $T$ for $t \in (0, 1)$ conditioned on $T$'s values $x_0, x_1$ at $t = 0, 1$.

Let $W^\epsilon \in \mathcal{P}(\Omega)$ denote the Wiener process with volatility $\epsilon > 0$ which starts at $p_0$ at $t = 0$. Its differential satisfies the stochastic differential equation (SDE): $dW_t^\epsilon = \sqrt{\epsilon}dW_t$. The SB problem with the Wiener prior $W^\epsilon$ between $p_0, p_1$ is to minimize the following objective:

$$\min_{T \in \mathcal{F}(p_0,p_1)} \mathrm{KL}\left(T \| W^\epsilon\right),\tag{2}$$

where $\mathcal{F}(p_0, p_1) \subset \mathcal{P}(\Omega)$ is the subset of stochastic processes which start at distribution $p_0$ (at the time $t = 0$) and end at $p_1$ (at $t = 1$). This problem has the unique solution which is a diffusion process $T^*$ described by the SDE: $dZ_t = g^*(Z_t, t)dt + dW_t^\epsilon$ (Léonard, 2013, Prop. 2.3). The process $T^*$ is called *the Schrödinger Bridge* and $g^* : \mathbb{R}^D \times [0, 1] \rightarrow \mathbb{R}^D$ is called *the optimal drift*.

**Equivalence between EOT and SB.** It is known that the solutions of EOT and SB are related to each other: for the EOT plan $\pi^*$ and the SB process $T^*$ it holds that $\pi^* = \pi^{T^*}$. Hence, solution $\pi^*$ to EOT (1) can be viewed as *a part of* the solution $T^*$ to SB (2). What *remains uncovered* by EOT in SB is the conditional process $T_{|x_0,x_1}$. Fortunately, it is known that $T_{|x_0,x_1} = W^\epsilon_{|x_0,x_1}$, i.e., it simply matches the "inner part" of the Wiener prior $W^\epsilon$ which is the well-studied Brownian Bridge (Pinsky & Karlin, 2011, Sec. 8.3.3). Hence, one may treat EOT and SB as equivalent problems.

**Characterization of solutions** (Léonard, 2013). The EOT plan $\pi^* = \pi^{T^*}$ has a specific form

$$\pi^*(x_0, x_1) = \psi^*(x_0) \exp\left(-\|x_0 - x_1\|^2/2\epsilon\right)\phi^*(x_1),\tag{3}$$

where $\psi^*, \phi^* : \mathbb{R}^D \rightarrow \mathbb{R}_+$ are two measurable functions called the *Schrödinger potentials*. They are defined up to multiplicative constants. The optimal drift $g^*$ of SB can be derived from $\phi^*$:

$$g^*(x, t) = \epsilon \nabla_x \log \int_{\mathbb{R}^D} \mathcal{N}(x'|x, (1-t)\epsilon I_D)\phi^*(x')dx'.\tag{4}$$

To use (4), it suffices to know $\phi^*$ up to the multiplicative constant.

**Computational Schrödinger Bridge setup.** Even though SB/EOT have many useful theoretical properties, solving the SB/EOT problems remains challenging in practice. Analytical solution is available only for the Gaussian case (Chen et al., 2015; Janati et al., 2020; Mallasto et al., 2022; Bunne et al., 2023) plus for some manually constructed benchmark pairs of distributions $p_0, p_1$ (Gushchin et al., 2023b). Moreover, in real world setting where SBs are applied (Vargas et al., 2021; Bunne et al., 2023; Tong et al., 2023), distributions $p_0$ and $p_1$ are almost never available explicitly but only through their *empirical samples*. For the rigor of the exposition, below we formalize the typical EOT/SB **learning setup** which we consider in our paper.

> We assume that the learner has access to empirical samples $X_0^N = \{x_0^1, x_0^2, \ldots, x_0^N\} \sim p_0$ and $X_1^M = \{x_1^1, x_1^2, \ldots, x_1^M\} \sim p_1$ from the (unknown) data distributions $p_0, p_1 \in \mathcal{P}_{2,ac}(\mathbb{R}^D)$. These samples (*train* data) are assumed to be independent. The task of the learner is to recover the solution (process $T^*$ or plan $\pi^*$) to SB problem (2) between the entire underlying distributions $p_0, p_1$.

The setup above is the learning from empirical samples and is usually called the **continuous OT**. In the continuous setup, it is essential to be able to do the *out-of sample estimation*, e.g., simulate the SB process trajectories starting from new (*test*) points $x_0^{new} \sim p_0$ and ending at $p_1$. In some practical use cases of SB, e.g., generative modeling (De Bortoli et al., 2021) and data-to-data translation (Gushchin et al., 2023a), a user is primarily interested only in the ends of these trajectories, i.e., new synthetic data samples from $p_1$. In the biological applications, it may be useful to study the entire trajectory as well (Bunne et al., 2023; Pariset et al., 2023). Hence, finding the solution to SB usually implies recovering the drift $g^*$ of SB or the conditional distributions $\pi^*(x_1|x_0)$ of the EOT plan.

## 3 LIGHT SCHRÖDINGER BRIDGE SOLVER

In §3.1, we derive the learning objective for our light SB solver. In §3.2, we present its training and inference procedures. In §3.3, we prove that our solver is a universal approximator of SBs.

### 3.1 DERIVING THE LEARNING OBJECTIVE

The main idea of our solver is to recover a parametric approximation $\pi_\theta \approx \pi^*$ of the EOT plan. Then this learned plan $\pi_\theta$ will be used to construct an approximation $T_\theta \approx T^*$ to the entire SB process (§3.2). To learn $\pi_\theta$, we want to directly minimize the KL divergence between $\pi^*$ and $\pi_\theta$:

$$\mathrm{KL}\left(\pi^* \| \pi_\theta\right) \to \min_{\theta \in \Theta}. \tag{5}$$

This objective is straightforward but there is an obvious obstacle: we do not know the EOT plan $\pi^*$. The magic is that optimization (5) can still be performed despite not knowing $\pi^*$. To begin with, recall that we already know that the EOT plan $\pi^*$ has a specific form (3). We define $u^*(x_0) \stackrel{\text{def}}{=} \exp(-\frac{\|x_0\|^2}{2\epsilon})\psi^*(x_0)$ and $v^*(x_1) \stackrel{\text{def}}{=} \exp(-\frac{\|x_1\|^2}{2\epsilon})\phi^*(x_1)$. These two are measurable functions $\mathbb{R}^D \to \mathbb{R}_+$, and we call them the *adjusted* Schrödinger potentials. Now (3) reads as

$$\pi^*(x_0, x_1) = u^*(x_0)\exp\left(\langle x_0, x_1\rangle/\epsilon\right)v^*(x_1) \quad \Rightarrow \quad \pi^*(x_1|x_0) \propto \exp\left(\langle x_0, x_1\rangle/\epsilon\right)v^*(x_1). \tag{6}$$

Our idea is to exploit this knowledge to parameterize $\pi_\theta$. We define

$$\pi_\theta(x_0, x_1) = p_0(x_0)\pi_\theta(x_1|x_0) = p_0(x_0)\frac{\exp\left(\langle x_0, x_1\rangle/\epsilon\right)v_\theta(x_1)}{c_\theta(x_0)}, \tag{7}$$

i.e., we parameterize $v^*$ as $v_\theta$. In turn, $c_\theta(x_0) \stackrel{\text{def}}{=} \int_{\mathbb{R}^D} \exp\left(\langle x_0, x_1\rangle/\epsilon\right)v_\theta(x_1)dx_1$ is the normalization.

Our parameterization guarantees that $\int_{\mathbb{R}^D} \pi_\theta(x_0, x_1)dx_1 = p_0(x_0)$. This is reasonable as in practice a learner is interested in the conditional distributions $\pi^*(x_1|x_0)$ rather than the density $\pi^*(x_0, x_1)$ of the plan. To be precise, we parameterize all the conditional distributions $\pi_\theta(x_1|x_0)$ via a common potential $v_\theta$. Below we will see that it is sufficient for both training and inference. In Appendix E, we show that within our framework it is easy to also parameterize the density of $p_0$ in (7). Surprisingly, this approach naturally coincides with just fitting a separate density model for $p_0$.

Now we show that optimization (5) can be performed without the knowledge of $\pi^*$.

**Proposition 3.1** (Feasible reformulation of the KL minimization). *For parameterization (7), it holds that the main KL objective (5) admits the representation* $\mathrm{KL}\left(\pi^* \| \pi^\theta\right) = \mathcal{L}(\theta) - \mathcal{L}^*$, *where*

$$\mathcal{L}(\theta) \stackrel{\text{def}}{=} \int_{\mathbb{R}^D} \log c_\theta(x_0)p_0(x_0)dx_0 - \int_{\mathbb{R}^D} \log v_\theta(x_1)p_1(x_1)dx_1, \tag{8}$$

*and* $\mathcal{L}^* \in \mathbb{R}$ *is a constant depending on distributions* $p_0, p_1$ *and value* $\epsilon > 0$ *but not on* $\theta$.

We see that minimizing KL equals to the minimization of the difference in expectations of $\log c_\theta(x_0)$ and $\log v_\theta(x_1)$ w.r.t. $p_0, p_1$, respectively. This means that the objective value (8) admits Monte-Carlo estimation from random samples and (8) can be optimized by using the stochastic gradient descent w.r.t. $\theta$. Yet there is still an obstacle that $c_\theta$ may be hard to compute analytically. We fix this below.

We recall (6) with $x_0 = 0$ and see that $\pi^*(x_1|x_0 = 0) \propto v^*(x_1)$, i.e., $v^*$ can be viewed as an unnormalized density of a distribution. Inspired by this observation, we employ the (unnormalized) Gaussian mixture parameterization for the adjusted potential $v_\theta$:

$$v_\theta(x_1) \stackrel{\text{def}}{=} \sum_{k=1}^{K} \alpha_k \mathcal{N}(x_1|r_k, \epsilon S_k), \tag{9}$$

where $\theta \stackrel{\text{def}}{=} \{\alpha_k, r_k, S_k\}_{k=1}^K$ are the parameters: $\alpha_k \geq 0$, $r_k \in \mathbb{R}^D$ and symmetric $0 \prec S_k \in \mathbb{R}^{D \times D}$. Note that we multiply $S_k$ in (9) by $\epsilon$ just to simplify the further derivations. For parameterization (9), conditional distributions $\pi_\theta(x_1|x_0)$ and normalization constants $c_\theta(x_0)$ are tractable.

**Proposition 3.2** (Tractable form of plan's components). *For the Gaussian Mixture parameterization (9) of the adjusted Schrödinger potential* $v_\theta$ *in (7), it holds that*

$$\pi_\theta(x_1|x_0) = \frac{1}{c_\theta(x_0)}\sum_{k=1}^{K} \widetilde{\alpha}_k(x_0)\mathcal{N}(x_1|r_k(x_0), \epsilon S_k) \qquad where \qquad r_k(x_0) \stackrel{\text{def}}{=} r_k + S_k x_0,$$

$$\widetilde{\alpha}_k(x_0) \stackrel{\text{def}}{=} \alpha_k \exp\left(\frac{x_0^T S_k x_0 + 2r_k^T x_0}{2\epsilon}\right), \qquad c_\theta(x_0) \stackrel{\text{def}}{=} \sum_{k=1}^{K} \widetilde{\alpha}_k(x_0).$$

The proposition provides the closed-form expression for $c_\theta(x_0)$ which is needed to optimize (8).
**Relation to prior work.** Optimizing objective (8) and using the Gaussian mixture parameterization (9) for the Schrödinger potentials are two ideas that appeared separately in (Mokrov et al., 2024) and (Gushchin et al., 2023b), respectively, and in different contexts. Our contribution is to combine these ideas together to get an efficient and light solver to SB, see the details below.

**(a) Energy-based view on EOT.** In (Mokrov et al., 2024), the authors aim to solve the continuous EOT problem. Using the duality for weak OT, the authors derive the objective which matches our (8) up to some change of variables, see their eq. (17) and Theorem 2. The lack of closed form for the normalization constant forces the authors to use the complex energy-based modeling (LeCun et al., 2006; Song & Kingma, 2021, EBM) techniques to perform the optimization. This is computationally heavy due to using the MCMC both during training and inference of the solver. Compared to their work, we employ a special parameterization of the Schrödinger potential, which allows to obtain the closed form expression for the normalizing constant. In turn, this allows us to optimize (8) directly with the minibatch gradient descent and *removes the burden of using MCMC at any point*. Besides, our solver yields the *closed form expression for SB* (see §3.2) while their approach does not.

**(b) Parameterization of Schrödinger potentials with Gaussian mixtures.** In (Gushchin et al., 2023a), the authors are interested in manually constructing pairs of continuous probability distributions for which the ground truth EOT/SB solution is available analytically. They propose a generic method to do this (Theorem 3.2) and construct several pairs to be used as a benchmark for EOT/SB solvers. The key building block in their methodology is to use a kind of the Gaussian mixture parameterization for Schrödinger potentials, which allows to obtain the closed form EOT and SB solutions for the constructed pairs (Proposition 3.3, Corollary 3.5). Our parameterization (9) coincides with theirs up to some change of variables. The important difference is that we use it to *learn the EOT/SB solution* (via learning parameters $\theta$ by optimizing (8) for a given pair of distributions $p_0, p_1$. At the same time, the authors pick the parameters $\theta$ at random to simply set up some potential and use it to build some pairs of distributions with the EOT plan between them available by their construction.

To summarize, our contribution is to unite these two separate ideas **(a)** and **(b)** from the field of EOT/SB. We obtain a straightforward minimization objective (8) which can be easily approached by standard gradient descent (§3.2). This makes the process of solving EOT/SB easier and faster.

## 3.2 TRAINING AND INFERENCE PROCEDURES

TRAINING. As the distributions $p_0, p_1$ are accessible only via samples $X^0 = \{x_0^1, \ldots, x_0^N\} \sim p_0$ and $X^1 = \{x_1^1, \ldots, x_1^M\} \sim p_1$ (recall the setup in §2), we optimize the empirical counterpart of (8):[1]

$$\widehat{\mathcal{L}}(\theta) \stackrel{\text{def}}{=} \frac{1}{N} \sum_{n=1}^{N} \log c_\theta(x_0^n) - \frac{1}{M} \sum_{m=1}^{M} \log v_\theta(x_1^m) \approx \mathcal{L}(\theta). \tag{10}$$

We use the (minibatch) gradient descent w.r.t. parameters $\theta$. To further simplify the optimization, we consider **diagonal** matrices $S_k$ in our parameterization (9) of $v_\theta$. Not only does it help to drastically reduce the number of learnable parameters in $\theta$ but it also allows to quickly compute $S_k^{-1}$ in $O(D)$ time. This simplification strategy works reasonably well in practice, see §5 below. Importantly, it is theoretically justified: in fact, it suffices to even use scalar covariance matrices $S_k = \lambda_k I_D \succ 0$ in $v_\theta$, see our Theorem 3.4. The other details are in Appendix D.

INFERENCE. The conditional distributions $\pi_\theta(x_1|x_0)$ are mixtures of Gaussians whose parameters are given explicitly in Proposition 3.2. Hence, sampling $x_1$ given $x_0$ is straightforward and lightspeed. So far we have discussed EOT-related training and inference aspects and skipped the question how to use $\pi_\theta$ to set-up some process $T_\theta \approx T^*$ approximating SB. We fix it below.

With each distribution $\pi_\theta$ defined by (7) via formula (7), we associate the specific process $T = T_\theta$ whose joint distribution at $t = 0, 1$ matches $\pi_\theta$ and conditionals satisfy $T_{\theta|x_0,x_1} = W_{|0,1}^\epsilon$. Informally, this means that we "insert" the Brownian Bridge "inside" the joint distribution $\pi_\theta$ at $t = 0, 1$. Below we show that this process admits the closed-form drift $g_\theta$ and the quality of approximation of $T^*$ by $T_\theta$ is the same as that of approximation of $\pi^*$ by $\pi_\theta$.

**Proposition 3.3** (Properties of $T_\theta$)**.** *Let $v_\theta$ be an unnormalized Gaussian mixture given by (9) and $\pi_\theta$ given by (7). Then $T_\theta$ introduced above is a diffusion process governed by the following SDE:*

$$T_\theta \quad : \quad dX_t = g_\theta(X_t, t)dt + \sqrt{\epsilon}dW_t, \qquad X_0 \sim p_0, \tag{11}$$

---

[1] We discuss the generalization properties of our light SB solver in Appendix A.

$$g_\theta(x,t) \stackrel{def}{=} \epsilon\nabla_x \log\left(\mathcal{N}(x|0,\epsilon(1-t)I_D)\sum_{k=1}^{K}\left\{\alpha_k\mathcal{N}(r_k|0,\epsilon S_k)\mathcal{N}(h(x,t)|0,A_k^t)\right\}\right)$$

*with $A_k^t \stackrel{def}{=} \frac{t}{\epsilon(1-t)}I_D + \frac{S_k^{-1}}{\epsilon}$ and $h_k(x,t) \stackrel{def}{=} \frac{1}{\epsilon(1-t)}x + \frac{1}{\epsilon}S_k^{-1}r_k$. Moreover, it holds that*

$$KL\left(T^*\|T_\theta\right) = KL\left(\pi^*\|\pi_\theta\right). \tag{12}$$

The proposition provides a closed form for the drift $g_\theta$ of $T_\theta$ for all $(x,t) \in \mathbb{R}^D \times [0,1]$. Now that we know what the process looks like, it is straightforward to sample its random trajectories starting at given input points $x_0$. We describe two ways for it based on the well-known schemes.

**Euler-Maryama simulation**. This is the well-celebrated time-discretized scheme to solve SDEs. Let $\Delta t = \frac{1}{S}$ be the time discretization step for an integer $S > 0$. Consider the following iteratively constructed (for $s \in \{0, 1, S-1\}$) sequence starting at $x_0$:

$$x_{(s+1)\Delta t} \leftarrow x_{s\Delta t} + g_\theta(x_{s\Delta t}, s\Delta_t)\Delta t + \sqrt{\epsilon\Delta_t}\xi_s \quad \text{with} \quad \xi_s \sim \mathcal{N}(0, I), \tag{13}$$

where $\xi_s$ are i.i.d. random Gaussian variables. Then the sequence $\{x_{s\Delta t}\}_{s=1}^{S}$ is a time-discretized approximation of some true trajectory of $T_\theta$ starting from $x_0$. Since our solver provides closed form $g_\theta$ (Proposition 3.3) for all $t \in [0,1]$, one may employ any arbitrary small discretization step $\Delta t$.

**Brownian Bridge simulation.** Given a start point $x_0$, one can sample an endpoint $x_1 \sim \pi_\theta(x_1|x_0)$ from the respective Gaussian mixture (Proposition 3.2). What remains is to sample the trajectory from the conditional process $T_{\theta|x_0,x_1}$ which matches the Brownian Bridge $W_{|x_0,x_1}^{\epsilon}$. Suppose that we already have some trajectory $x_0, x_{t_1}, \ldots, x_{t_L}, x_1$ with $0 < t_1 < \cdots < t_L < 1$ (initially $L = 0$, we have only $x_0, x_1$), and we want to refine the trajectory by inserting a point at $t_l < t < t_{l+1}$. Following the properties of the Brownian bridge, it suffices to sample

$$x_t \sim \mathcal{N}\left(x_t|x_{t_l} + \frac{t'-t_l}{t_{l+1}-t_l}(x_{t_{l+1}} - x_{t_l}), \epsilon\frac{(t'-t_l)(t_{l+1}-t')}{t_{l+1}-t_l}\right). \tag{14}$$

Using this approach, one may sample arbitrarily precise trajectories of $T_\theta$ *without* any discretization errors. Unlike (13), this sampling technique does not use the drift $g_\theta$ and to get a random sample at any time $t$ one does not need to sequentially unroll the entire prior trajectory at $[0, t)$.

### 3.3 Universal Approximation Property

Considering plans $\pi_\theta$ with the Gaussian mixture parameterization (9), it is natural to wonder whether this parameterization is universal. Namely, we aim to understand whether $T_\theta$ can approximate any Schrödinger bridge arbitrarily well if given a sufficient amount of components in the mixture $v_\theta$. We provide a positive answer assuming that $p_0, p_1$ are supported on compact sets. This assumption is not restrictive as in practice many real-world distributions are compactly supported anyway.

**Theorem 3.4** (Gaussian mixture parameterization for the adjusted potential provides the universal approximation of Schrödinger bridges). *Assume that $p_0$ and $p_1$ are compactly supported. Then for all $\delta > 0$ there exists a Gaussian mixture $v_\theta$ (9) with **scalar** covariances $S_k = \lambda_k I_D \succ 0$ of its components that satisfies the inequality $KL\left(T^*\|T_\theta\right) = KL\left(\pi^*\|\pi_\theta\right) < \delta$.*

Although this result looks concise and simple, its proof is quite **challenging**. The main cornerstone in proving the result is that the key object to be approximated, i.e., the adjusted potential $v^*$, is just a measurable function without any nice properties such as the continuity. To overcome this issue, we employ non-trivial facts from the duality theory for weak OT (Gozlan et al., 2017).

We also highlight the fact that our result provides an approximation of $T^*$ on the **non-compact** set. Indeed, while $p_0$ and $p_1$ are compactly supported, $T_\theta$'s marginals at all the time steps $t \in (0,1)$ are always supported on the entire $\mathbb{R}^D$ which is not compact, recall, e.g., (14). This aspect adds additional value to our result as usually the approximation is studied in the compact sets. To our knowledge, our Theorem 3.4 is the first ever result about the universal approximation of SBs.

## 4 Related Work

Over several recent years, there has been a notable progress in developing neural SB/EOT solvers. For a review and a benchmark of them, we refer to (Gushchin et al., 2023b). The dominant majority of them have rather non-trivial training or inference procedures, which complicates the usage of them in practical problems. Below we summarize their main principles.

| Solver \ Property | Allows to sample from $\pi^*(\cdot\|x)$ | Non-minimax objective | Non-iterative objective | Non-simulation based training | Recovers the drift $g^*(x,t)$ | Recovers the density of $\pi^*(\cdot\|x)$ | Does not use simulation inference | Satisfies the universal approximation | Works for reasonably small $\epsilon$ |
|---|---|---|---|---|---|---|---|---|---|
| (Seguy et al., 2018) | ✗ | ✓ | ✓ | ✓ | ✗ | ✗ | ✓ | ? | ✗ |
| (Daniels et al., 2021) | ✓ | ✓ | ✓ | ✓ | ✗ | ✗ | ✗ | ? | ✗ |
| (Mokrov et al., 2024) | ✓ | ✓ | ✓ | ✗ | ✗ | ✗ | ✗ | ? | ✓ |
| (Gushchin et al., 2023a) | ✓ | ✗ | ✓ | ✗ | ✓ | ✗ | ✗ | ? | ✓ |
| (Vargas et al., 2021) | ✓ | ✓ | ✗ | ✗ | ✓ | ✗ | ✗ | ? | ✓ |
| (De Bortoli et al., 2021) | ✓ | ✓ | ✗ | ✗ | ✓ | ✗ | ✗ | ? | ✓ |
| (Chen et al., 2021a) | ✓ | ✓ | ✗ | ✗ | ✓ | ✗ | ✗ | ? | ✓ |
| (Shi et al., 2023) | ✓ | ✓ | ✗ | ✗ | ✓ | ✗ | ✗ | ? | ✓ |
| (Kim et al., 2024) | ✓ | ✗ | ✓ | ✗ | ✓ | ✗ | ✗ | ? | ✓ |
| (Tong et al., 2023) | ✓ | ✓ | ✓ | ✓ | ✓ | ✗ | ✗ | ? | ✓ |
| LightSB (**ours**) | ✓ | ✓ | ✓ | ✓ | ✓ | ✓ | ✓ | ✓ | ✓ |

Table 1: Comparison of features of existing EOT/SB solvers and **our** proposed light solver.

**Dual form solvers for EOT.** The works (Genevay et al., 2016; Seguy et al., 2018; Daniels et al., 2021) aim to solve EOT (1), i.e., the static version of SB. The authors approach the classic dual EOT problem (Genevay, 2019, §3.1) with neural networks. They recover the conditional distributions $\pi^*(x_1|x_0)$ or only the barycentric projections $x_0 \mapsto \int_{\mathbb{R}^D} x_1 \pi^*(x_1|x_0) dx_1$ without learning the actual SB process $T^*$. Unfortunately, both solvers do not work for small $\epsilon$ due to numerical instabilities (Gushchin et al., 2023b, Table 2). At the same time, large $\epsilon$ is of limited practical interest as the EOT plan is nearly independent, i.e., $\pi^*(x_0, x_1) \approx p_0(x_0)p_1(x_1)$ and $\pi^*(x_1|x_0) \approx p_1(x_1)$. In this case, it becomes reasonable just to learn the unconditional generative model for $p_1$. This issue is addressed in the work (Mokrov et al., 2024) which we discussed in §3.1. There the authors consider the weak OT dual form (Backhoff-Veraguas et al., 2019, Theorem 1.3) for EOT and demonstrate that it can be approached with energy-based modeling techniques (LeCun et al., 2006, EBM). Their solver as well as (Daniels et al., 2021) still heavily rely on using time-consuming MCMC techniques.

The above-mentioned solvers can be also adapted to sample trajectories from SB by using the Brownian Bridge just as we do in §3.2. However, unlike our light solver, these solvers do not provide an access to the optimal drift $g^*$. Recently, (Gushchin et al., 2023a) demonstrated that one may reformulate the weak EOT dual problem so that one can get the SB's drift $g^*$ from its solution as well. However, their solver requires dealing with a challenging max-min optimization problem and requires simulating the full trajectories of the learned process, which complicates training.

**Iterative proportional fitting (IPF) solvers for SB.** Most SB solvers (Vargas et al., 2021; De Bortoli et al., 2021; Chen et al., 2021a; 2023) directly aim to recover the optimal drift $g^*$ as it can be later used to simulate the SB trajectories as we discussed in §3.2. Such solvers are mostly based on the iterative proportional fitting procedure (Fortet, 1940; Kullback, 1968; Ruschendorf, 1995), which is also known as the Sinkhorn algorithm (Cuturi, 2013) and, in fact, coincides with the well-known expectation-maximization algorithm (Dempster et al., 1977, EM), see (Vargas & Nüsken, 2023, Proposition 4.1) for discussion. That is, the above-mentioned solvers learn two SDEs (forward and inverse processes) and iteratively update them one after the other (IPF steps). The first two solvers do this via the mean-matching regression while the others optimize a divergence-based objective, see (Chen et al., 2021a, §1), (Chen et al., 2023, §5). All these solvers require performing multiple IPF steps. At each of the steps, they simulate the full trajectories of the learned process which introduces considerable computational overhead since these processes are represented via large neural networks. In particular, due to the error accumulation as IPF proceeds, it is known that such solvers may forget the Wiener prior, see (Vargas & Nüsken, 2023, §4.1.2) and references therein.

**Other solvers for EOT/SB**. In (Shi et al., 2023), a new approach to SB based on alternative Markovian and Reciprocal projections is introduced. In (Tong et al., 2023), the authors exploit the property that SB solution $T^*$ consists of entropic OT plan $\pi^*$ and Brownian bridge $W^\epsilon_{|x_0,x_1}$. They propose a

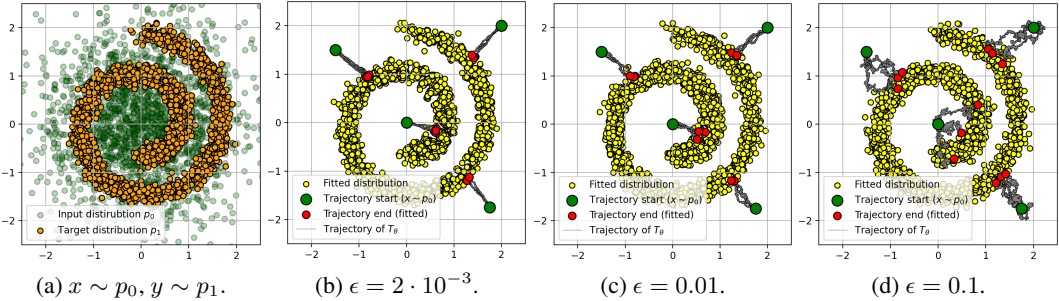

(a) $x \sim p_0$, $y \sim p_1$.     (b) $\epsilon = 2 \cdot 10^{-3}$.     (c) $\epsilon = 0.01$.     (d) $\epsilon = 0.1$.

Figure 2: The process $T_\theta$ learned with LightSB (**ours**) in *Gaussian→Swiss roll* example.

| | $\epsilon = 0.1$ | | | | $\epsilon = 1$ | | | | $\epsilon = 10$ | | | |
|---|---|---|---|---|---|---|---|---|---|---|---|---|
| | $D=2$ | $D=16$ | $D=64$ | $D=128$ | $D=2$ | $D=16$ | $D=64$ | $D=128$ | $D=2$ | $D=16$ | $D=64$ | $D=128$ |
| Best solver | 1.94 | 13.67 | 11.74 | 11.4 | 1.04 | 9.08 | 18.05 | 15.23 | 1.40 | 1.27 | 2.36 | 1.31 |
| ⌊**LightSB**⌉ | **0.03** | **0.08** | **0.28** | **0.60** | **0.05** | **0.09** | **0.24** | **0.62** | **0.07** | **0.11** | **0.21** | **0.37** |
| $\pm std$ | ±0.01 | ±0.04 | ±0.02 | ±0.02 | ±0.003 | ±0.006 | ±0.007 | ±0.007 | ±0.02 | ±0.01 | ±0.01 | ±0.01 |

Table 2: Comparisons of $c\mathbb{BW}_2^2$-UVP $\downarrow$ (%) between the optimal plan $\pi^*$ and the learned plan $\pi_\theta$.

new objective to learn $T^*$ in the form of SDE if the EOT plan $\pi^*$ is known. Since $\pi^*$ is not actually known, the authors use the minibatch OT to approximate it. In (Kim et al., 2024), the authors propose exploiting the self-similarity of the SB problem and consider the family of SB problems on intervals $\{[t_i, 1]\}_{i=1}^N$ to sequentially learn $T^*$ as a series of conditional distributions $x_{t_{i+1}}|x_{t_i}$. However, they add an empirical regularization which may bias the solution. Besides, there exist SB solvers for specific setups with the paired train data available (Somnath et al., 2023; Liu et al., 2023).

**Summary.** We provide a Table 1 with the summary of the features of the discussed EOT/SB solvers. Additionally, in Appendix F, we mention other OT solvers which are related but not closely relevant to our paper because of considering non EOT/SB formulations or non-continuous settings.

## 5 EXPERIMENTAL ILLUSTRATIONS

Below we evaluate our light solver on setups with both synthetic (§5.1, §5.2) and real data distributions (§5.3, §5.4). The code for our solver is written in PyTorch available at https://github.com/ngushchin/LightSB. The experiments are issued in the form of convenient *.ipynb notebooks. **Reproducing each experiment requires a few minutes on CPU with 4 cores**. The implementation and experimental details are given in Appendix D.

### 5.1 TWO-DIMENSIONAL EXAMPLES

To show the effect of $\epsilon$ on the learned process $T_\theta$, we give a toy example of mapping 2D *Gaussian→Swiss Roll* with our light solver for $\epsilon = 2 \cdot 10^{-3}, 10^{-2}, 10^{-1}$, see Fig. 2. As expected, for small $\epsilon$ the trajectories are almost straight, and the process $T_\theta$ is nearly deterministic. The volatility of trajectories increases with $\epsilon$, and the conditional distributions $\pi_\theta(x_1|x_0)$ become more disperse.

### 5.2 EVALUATION ON THE EOT/SB BENCHMARK

To empirically verify that our light solver correctly recovers the EOT/SB, we evaluate it on a recent EOT/SB benchmark by (Gushchin et al., 2023b, §5). The authors provide high-dimensional continuous distributions $(p_0, p_1)$ for which the ground truth conditional EOT plan $\pi^*(\cdot|x_0)$ and SB process $T^*$ are known by the construction. Moreover, they use these pairs to evaluate many solvers from §4.

We use their *mixtures* benchmark pairs (see their §4) with various dimensions and $\epsilon$, and use the same conditional $\mathbb{BW}_2^2$-UVP metric (see their §5) to compare our recovered plan $\pi_\theta$ with the ground truth plan $\pi^*$. In Table 2, we report the results of our solver vs. the best solver in each setup according to their evaluation. As clearly seen, our solver outperforms the best solver by a *considerable* margin. This is reasonable as the benchmark distributions are constructed using the similar principles which our solver exploits, namely, the sum-exp (Gaussian mixture) parameterization of the Schrödinger potential. Therefore, our light solver has a considerable inductive bias for solving the benchmark.

### 5.3 SINGLE CELL DATA

One of the important applications of SB is the analysis of biological single cell data (Koshizuka & Sato, 2022; Bunne et al., 2021; 2022). In Appendix C, we evaluate our algorithm on the popular embryonic stem cell differentiation dataset which has been used in many previous works (Tong et al., 2020; Vargas et al., 2021; Bunne et al., 2023; Tong et al., 2023); here we consider the more high-dimensional dataset from the Kaggle completion "Open Problems - Multimodal Single-Cell Integration" (MSCI) which was first used in (Tong et al., 2023). The MSCI dataset consists of single-cell data from four human donors at 4 time points (days 2, 3, 4, and 7). We solve the SB/EOT problem between distribution pairs at days 2 and 4, 3 and 7, and evaluate how well the solvers recover the intermediate distributions at days 3 and 4 correspondingly. We work with PCA projections

| Setup | Solver type | DIM / Solver | 50 | 100 | 1000 |
|---|---|---|---|---|---|
| Discrete EOT | Sinkhorn | (Cuturi, 2013) [1 GPU V100] | 2.34 (90 s) | 2.24 (2.5 m) | 1.864 (9 m) |
| Continuous EOT | Langevin-based | (Mokrov et al., 2024) [1 GPU V100] | $2.39 \pm 0.06$ (19 m) | $2.32 \pm 0.15$ (19 m) | $1.46 \pm 0.20$ (15 m) |
| Continuous EOT | Minimax | (Gushchin et al., 2023a) [1 GPU V100] | $2.44 \pm 0.13$ (43 m) | $2.24 \pm 0.13$ (45 m) | $1.32 \pm 0.06$ (71 m) |
| Continuous EOT | IPF | (Vargas et al., 2021) [1 GPU V100] | $3.14 \pm 0.27$ (8 m) | $2.86 \pm 0.26$ (8 m) | $2.05 \pm 0.19$ (11 m) |
| Continuous EOT | KL minimization | LightSB (**ours**) [4 CPU cores] | $2.31 \pm 0.27$ (65 s) | $2.16 \pm 0.26$ (66 s) | $1.27 \pm 0.19$ (146 s) |

Table 3: Energy distance (averaged for two setups and 5 random seeds) on the MSCI dataset along with 95%-confidence interval ($\pm$ intervals) and average training times (s - seconds, m - minutes).

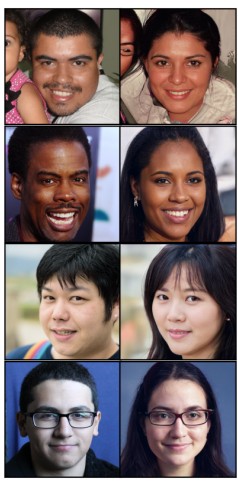 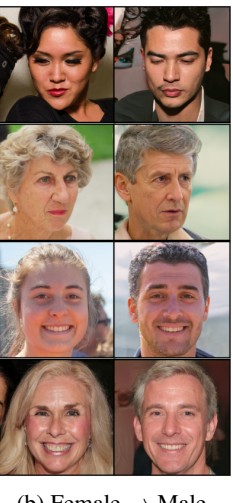 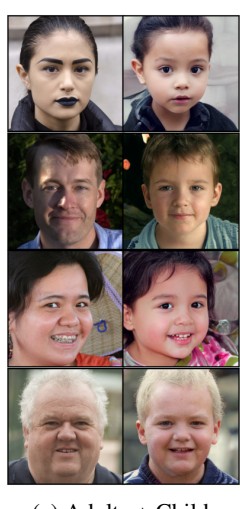 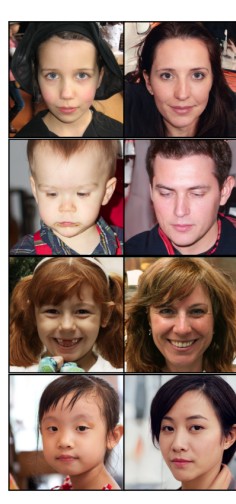

(a) Male → Female.  (b) Female → Male.  (c) Adult → Child.  (d) Child → Adult.

Figure 3: Unpaired translation by our LightSB solver applied in the latent space of ALAE for 1024x1024 FFHQ images. *Our LightSB converges on 4 cpu cores in less than 1 minute.*

with DIM $= 50, 100, 1000$ components. We use the energy distance (Rizzo & Székely, 2016, ED) as a metric and present the results for different classes of SB solvers in Table 3 along with the training time. We see that LightSB achieves similar quality to other EOT/SB solvers but faster and without GPU. The underline{details} of used preprocessing, hyperparameter and baselines are in Appendix D.4

### 5.4 Unpaired Image-to-image Translation

One application which is frequently considered in EOT/SB papers (Daniels et al., 2021; Chen et al., 2021b) is the unpaired image-to-image translation (Zhu et al., 2017). Our solver may be hard to apply to learning SB directly in the image space. To be precise, it is *not designed* to be used in image spaces just like the conventional Gaussian mixture model is not used for image synthesis.

Still we show that our solver might be handy for working in the latent spaces of generative models. We consider the task of *male→female* translation. We pick pre-trained ALAE autoencoder (Pidhorskyi et al., 2020) for entire $1024 \times 1024$ FFHQ dataset (Karras et al., 2019) of 70K human faces. We split first 60K faces (train) into *male* and *female* subsets and use the encoder to extract 512-dimensional latent codes $\{z_0^n\}_{n=1}^N$ and $\{z_1^m\}_{m=1}^M$ from the images in each subset.

**Training**. We learn the latent EOT plan $\pi_\theta(z_1|z_0)$ by using the above-mentioned unpaired samples from the latent distributions. *The training process takes less than 1 minute on 4 CPU cores.*

**Inference.** To perform *male→female* translation for a new *male* face $x_0^{new}$ (from 10K test faces), we **(1)** encode it via $z_0^{new} = \text{Enc}(x_0^{new})$, **(2)** sample $z_1 \sim \pi_\theta(z_1|z_0^{new})$ and then **(3)** decode $x_1 = \text{Dec}(z_1)$ and return it. Note that here (unlike §5.3) the process $T_\theta$ is not needed, only $\pi_\theta$ is used.

**Results.** The qualitative results are given in Fig. 1 and Fig. 3a. Furthermore, in Fig. 3, we provide additional examples for other setups: *female→male* and *child↔adult*. For brevity, we show only 1 translated images per an input image. In Appendix H, we give extra examples and study the effect of $\epsilon$. Our experiments qualitatively confirm that our LightSB can solve distribution translation tasks in high dimensions ($D = 512$), and it can be used to easily convert auto-encoders to translation models.

### 6 Discussion

**Potential impact**. Compared to the existing EOT/SB solvers, our light solver provides many advantages (Table 1). It is one-step (no IPF steps), does not require max-min optimization, does not require the simulation of the process during the training, provides the closed form of the learned drift $g_\theta$ of the process $T_\theta \approx T^*$ and of the conditional distributions $\pi_\theta(x_1|x_0) \approx \pi^*(x_1|x_0)$ of the plan. Moreover, our solver is provably a universal approximator of SBs. Still the **key benefit** of our light solver is its simplicity and ease to use: it has a straightforward optimization objective and does not use heavy-weighted neural parameterization. These facts help our light solver to converge in a matter of minutes without spending a lot of user/researcher's time on setting up dozens of hyperparameters, e.g., neural network architectures, and waiting hours for the solver to converge on GPUs. We believe that these properties could help our solver to become the standard easy-to-use baseline EOT/SB solver with potential applications to data analysis tasks.

Limitations and broader impact are discussed in Appendix G.

## 7 REPRODUCIBILITY

The code for our solver is available at



`https://github.com/ngushchin/LightSB.`



1. To reproduce experiments from §5.1 it is enough to train LightSB model by running notebook `notebooks/LightSB_swiss_roll.ipynb` with hyperparameters described in §D.2 and then run notebook `notebooks/swiss_roll_plot.ipynb` to plot Fig. 2.

2. To reproduce experiments from §5.2 it is needed to install Entropic OT benchmark from github `https://github.com/ngushchin/EntropicOTBenchmark` and then run notebook `LightSB_EOT_benchmark.ipynb` with hyperparameters described in §D.3 to reproduce reported metrics in Table 2.

3. To reproduce experiments from Appendix C it is needed to install library from `https://github.com/KrishnaswamyLab/TrajectoryNet` and then to run notebook `notebooks/LightSB_single_cell.ipynb`. All required data is already preprocessed and located in `data` folder.

4. To reproduce experiments from §5.3 it is needed to download data from `https://www.kaggle.com/competitions/open-problems-multimodal/` and then to run notebook `data/data_preprocessing.ipynb` to preprocess data. The experiments with LightSB solver can be reproduced by running the notebook `notebooks/LightSB_MSCI.ipynb`. The experiments with Sinkhorn solver can be reproduced by running the notebook `notebooks/Sinkhorn_MSCI.ipynb`

5. The code for ALAE is already included in our code and to reproduce experiments from §5.4 it is first necessary to load the ALAE model by running the script `ALAE/training_artifacts/download_all.py`. We have already coded the FFHQ dataset from ALAE and these data can be downloaded directly using notebook `notebooks/LightSB_alae.ipynb`. To train the LightSB model it is necessary to run the notebook `notebooks/LightSB_alae.ipynb`. The same notebook also contains code for plotting the results of trained models.

## 8 ACKNOWLEDGEMENTS

The work was supported by the Analytical center under the RF Government (subsidy agreement 000000D730321P5Q0002, Grant No. 70-2021-00145 02.11.2021).

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

## A   GENERALIZATION PROPERTIES OF OUR LIGHT SOLVER

In theory, to recover the optimal plan $\pi^*$ one can solve $\mathcal{L}(\theta) \to \min_\theta$ which is equivalent to the direct minimization of KL $(\pi^*\|\pi_\theta)$ w.r.t. $\theta$ (Proposition 3.1). According to (8), $\mathcal{L}(\theta)$ consists of the difference of integrals of $\log c_\theta(x_0)$ and $\log v_\theta(x_1)$ over the distributions $p_0$ and $p_1$, respectively. In practice, there are several sources of errors which do not allow to perfectly optimize the objective.

1. **Statistical (estimation) error**. Since distributions $p_0, p_1$ are accessible only via empirical samples $X^0 = \{x_0^1, \ldots, x_0^N\} \sim p_0$ and $X^1 = \{x_1^1, \ldots, x_1^N\} \sim p_1$, one is forced to optimize the empirical counterpart $\widehat{\mathcal{L}}(\theta)$ of $\mathcal{L}(\theta)$. In this objective, the integrals over $p_0, p_1$ are replaced with their estimates using samples $X^0, X^1$, recall (10). For given samples $X^0, X^1$, we denote

$$\widehat{\theta} = \widehat{\theta}(X^0, X^1) = \arg\min_\theta \widehat{\mathcal{L}}(\theta). \tag{15}$$

   Usually, $\widehat{\mathcal{L}}(\theta)$ is called the *empirical risk* and $\widehat{\theta}$ is the *empirical risk minimizer*.

2. **Approximation error**. The parametric class for $v_\theta$ over which one optimizes the objective is restricted. For example, we consider (unnormalized) Gaussian mixtures $v_\theta$ with $K$ components (parametrized with $\theta = \{\alpha_k, r_k, S_k\}_{k=1}^K$). This may lead to irreducible error in approximation of the OT plan $\pi^*$ with $\pi_\theta$ due to parametric restrictions. In our setup, the quantity

$$\mathcal{L}(\theta^*) - \mathcal{L}^* = \min_\theta \mathcal{L}(\theta) - \mathcal{L}^* \tag{16}$$

   is the *approximation error*. Here $\theta^* = \arg\min_\theta \mathcal{L}(\theta)$ is the best approximator (in a given class).

3. **Optimization error**. In practice, we solve $\widehat{\mathcal{L}}(\theta) \to \min_\theta$ with the gradient descent. The optimization w.r.t. is non-convex and there are no general guarantees of convergence to the global empirical risk minimizer $\widehat{\theta}$. This may introduce an additional *optimization error*. Analysing this quantity is a too general question of the non-convex optimization and goes far beyond the scope of our paper. Therefore, for further analysis we assume this error to be zero.

Given the gap between the theoretical objective $\mathcal{L}(\theta)$ and its empirical counterpart $\widehat{\mathcal{L}}(\theta)$, it is natural to wonder how close is the recovered $\pi_{\widehat{\theta}}$ to $\pi^*$. We aim to obtain the bound for the expected KL between $\pi^*$ and $\pi_{\widehat{\theta}}$ (or, equivalently, $T^*$ and $T_\theta$), i.e., $\mathbb{E}\text{KL}\left(\pi^*\|\pi_{\widehat{\theta}}\right) = \mathbb{E}\text{KL}\left(T^*\|T_{\widehat{\theta}}\right)$, where the expectation is taken w.r.t. the random realization of the train data $X^0, X^1$. This quantity is the natural definition of the **generalization error** in our setting. Note that

$$\mathbb{E}\text{KL}\left(T^*\|T_{\widehat{\theta}}\right) \overset{\text{Prop. 3.3}}{=} \mathbb{E}\text{KL}\left(\pi^*\|\pi_{\widehat{\theta}}\right) \overset{\text{Prop. 3.1}}{=} \mathbb{E}\left[\mathcal{L}(\widehat{\theta}) - \mathcal{L}^*\right] =$$
$$\mathbb{E}\left[\mathcal{L}(\widehat{\theta}) - \mathcal{L}(\theta^*)\right] + \mathbb{E}\left[\mathcal{L}(\theta^*) - \mathcal{L}^*\right] = \underbrace{\mathbb{E}\left[\mathcal{L}(\widehat{\theta}) - \mathcal{L}(\theta^*)\right]}_{\text{Statistical error}} + \underbrace{\left[\mathcal{L}(\theta^*) - \mathcal{L}^*\right]}_{\text{Approximation error (16)}}. \tag{17}$$

Thanks to our Theorem 3.4, we already known that the second term (the approximation error) can be made arbitrarily small if we pick a Gaussian mixture with sufficiently large amount of components. Hence, our analysis below focuses on bounding the statistical error and understanding the rate of its convergence to zero as a function of available sample sizes $N, M$. Our following theorem demonstrates that the statistical error decreases at the usual **parametric** rate.

**Theorem A.1** (Bound for the statistical error). *Assume that $p_0, p_1$ are compactly supported. Assume that the considered parametric class $\Theta$ ($\ni \theta$) consists of (unnormalized) Gaussian mixtures with $K$ components with bounded means $\|r_k\| \leq R$ (for some $R > 0$), covariances $sI \preceq S_k \preceq SI$ (for some $0 < s \leq S$) and weights $a \leq \alpha_k \leq A$ (for some $0 < a \leq A$). Then the following holds:*

$$\mathbb{E}\left[\mathcal{L}(\widehat{\theta}) - \mathcal{L}(\theta^*)\right] \leq \frac{C_0}{\sqrt{N}} + \frac{C_1}{\sqrt{M}},$$

*where constants $C_0, C_1$ depend only on $K, R, s, S, a, A, p_0, p_1, \epsilon$ but not on sample sizes $M, N$.*

The proof is given in the next Appendix section. In the future, it would be interesting to study the trade-off between the statistical error and approximation error rather than study these instances separately as we do in our paper. However, providing such an analysis will probably require making stronger assumptions (e.g., smoothness) on the distributions $p_0, p_1$ and the true optimal adjusted Schrödinger potential $v^*$. We leave this interesting question for the future work.

## B  PROOFS

### B.1  PROOFS FOR THE RESULTS IN THE MAIN TEXT

*Proof of Proposition 3.1.*  In the derivations below, we use $H(\cdot)$ to denote the entropy, i.e., the minus KL divergence with the Legesgue measure. We obtain

$$\mathrm{KL}\left(\pi^* \| \pi_\theta\right) = \int_{\mathbb{R}^D \times \mathbb{R}^D} \pi^*(x_0, x_1) \log \frac{\pi^*(x_0, x_1)}{\pi_\theta(x_0, x_1)} dx_0 dx_1 =$$

$$-H(\pi^*) - \int_{\mathbb{R}^D \times \mathbb{R}^D} \pi^*(x_0, x_1) \log \big( \underbrace{p_0(x_0) \pi_\theta(x_1 | x_0)}_{=\pi_\theta(x_0, x_1)} \big) dx_0 dx_1 =$$

$$-H(\pi^*) - \int_{\mathbb{R}^D \times \mathbb{R}^D} \pi^*(x_0, x_1) \log p_0(x_0) dx_0 dx_1 - \int_{\mathbb{R}^D \times \mathbb{R}^D} \pi^*(x_0, x_1) \log \pi_\theta(x_1 | x_0) dx_0 dx_1 =$$

$$-H(\pi^*) - \underbrace{\int_{\mathbb{R}^D} \overbrace{\pi^*(x_0)}^{=p_0(x_0)} \log p_0(x_0) dx_0}_{=-H(p_0)} - \int_{\mathbb{R}^D \times \mathbb{R}^D} \pi^*(x_0, x_1) \log \frac{\exp\left(\langle x_0, x_1 \rangle / \epsilon\right) v_\theta(x_1)}{c_\theta(x_0)} dx_0 dx_1 =$$

$$\underbrace{-H(\pi^*) + H(p_0) - \epsilon^{-1} \int_{\mathbb{R}^D \times \mathbb{R}^D} \langle x_0, x_1 \rangle \pi^*(x_0, x_1) dx_0 dx_1}_{\stackrel{\text{def}}{=} -\mathcal{L}^*} + \quad (18)$$

$$\int_{\mathbb{R}^D \times \mathbb{R}^D} \pi^*(x_0, x_1) \log c_\theta(x_0) dx_0 dx_1 - \int_{\mathbb{R}^D \times \mathbb{R}^D} \pi^*(x_0, x_1) \log v_\theta(x_1) dx_0 dx_1 =$$

$$-\mathcal{L}^* + \int_{\mathbb{R}^D} p_0(x_0) \log c_\theta(x_0) dx_0 - \int_{\mathbb{R}^D} p_1(x_1) \log v_\theta(x_1) dx_1 = \mathcal{L}(\theta) - \mathcal{L}^*, \quad (19)$$

which is exactly what we need. $\qquad\qquad\square$

*Proof of Proposition 3.2.*  We use equation (7) for $\pi_\theta(x_0, x_1)$ and equation (9) for $v_\theta(x_1)$ to derive:

$$\pi_\theta(x_1 | x_0) = \frac{\exp\left(\langle x_0, x_1 \rangle / \epsilon\right) v_\theta(x_1)}{c_\theta(x_0)} = \frac{1}{c_\theta(x_0)} \exp\left(\langle x_0, x_1 \rangle / \epsilon\right) \sum_{k=1}^K \alpha_k \mathcal{N}(x_1 | r_k, \epsilon S_k) =$$

$$\frac{1}{c_\theta(x_0)} \sum_{k=1}^K \alpha_k \exp\left(\langle x_0, x_1 \rangle / \epsilon\right) \mathcal{N}(x_1 | r_k, \epsilon S_k) =$$

$$\frac{1}{c_\theta(x_0)} \sum_{k=1}^K \alpha_k (2\pi)^{-D/2} |\epsilon S_k|^{-1/2} \exp\left(\langle x_0, x_1 \rangle / \epsilon\right) \exp(-\frac{1}{2}(x_1 - r_k)^T \frac{S_k^{-1}}{\epsilon}(x_1 - r_k)) =$$

$$\frac{1}{c_\theta(x_0)} \sum_{k=1}^K \alpha_k (2\pi)^{-D/2} |\epsilon S_k|^{-1/2} \exp\left(\frac{1}{2\epsilon} \left(2x_0^T x_1 - (x_1 - r_k)^T S_k^{-1}(x_1 - r_k)\right)\right) =$$

$$\frac{1}{c_\theta(x_0)} \sum_{k=1}^K \alpha_k (2\pi)^{-D/2} |\epsilon S_k|^{-1/2} \exp\left(\frac{1}{2\epsilon}(2x_0^T x_1 - x_1^T S_k^{-1} x_1^T + 2r_k^T S_k^{-1} x_1 - r_k^T S_k^{-1} r_k)\right) =$$

$$\frac{1}{c_\theta(x_0)} \sum_{k=1}^K \alpha_k (2\pi)^{-D/2} |\epsilon S_k|^{-1/2} \exp\left(\frac{1}{2\epsilon}(-x_1^T S_k^{-1} x_1^T + 2 \underbrace{(S_k x_0 + r_k)^T}_{=r_k(x_0)} S_k^{-1} x_1 - r_k^T S_k^{-1} r_k)\right) =$$

$$\frac{1}{c_\theta(x_0)} \sum_{k=1}^K \alpha_k (2\pi)^{-D/2} |\epsilon S_k|^{-1/2} \exp\left(-\frac{1}{2\epsilon}(x_1 - r_k(x_0))^T S_k^{-1}(x_1 - r_k(x_0))\right.$$

$$\left. \exp\left(\frac{1}{2\epsilon}(-r_k^T S_k^{-1} r_k + r_k^T(x_0) S_k^{-1} r_k^T(x_0))\right) = \right.$$

$$\frac{1}{c_\theta(x_0)} \sum_{k=1}^K \alpha_k \exp\Big(\frac{-r_k^T S_k^{-1} r_k + r_k^T(x_0) S_k^{-1} r_k^T(x_0)}{2\epsilon}\Big) \mathcal{N}(x_1 | r_k(x_0), \epsilon S_k) =$$

$$\frac{1}{c_\theta(x_0)} \sum_{k=1}^K \alpha_k \exp\Big(\frac{r_k^T S_k^{-1} r_k + (S_k x_0 + r_k)^T(x_0) S_k^{-1}(S_k x_0 + r_k)(x_0)}{2\epsilon}\Big) \mathcal{N}(x_1 | r_k(x_0), \epsilon S_k) =$$

$$\frac{1}{c_\theta(x_0)} \sum_{k=1}^K \alpha_k \underbrace{\exp\Big(\frac{x_0^T S_k x_0 + 2 r_k^T x_0}{2\epsilon}\Big)}_{=\widetilde\alpha_k(x_0)} \mathcal{N}(x_1 | r_k(x_0), \epsilon S_k) =$$

$$\frac{1}{c_\theta(x_0)} \sum_{k=1}^K \widetilde\alpha_k(x_0) \mathcal{N}(x_1 | r_k(x_0), \epsilon S_k).$$

Since $\int_{\mathbb{R}^D} \pi_\theta(x_1|x_0) dx_1 = 1$, we see that $c_\theta = \sum_{k=1}^K \widetilde\alpha_k(x_0)$ and conclude the proof. $\qquad\square$

*Proof of Proposition 3.3.* Define $p_\theta = \int_{\mathbb{R}^D} \pi_\theta(x_0, x_1) dx_0$ as the density of the second marginal of $\pi_\theta$. From the OT benchmark constructor (Gushchin et al., 2023b, Theorem 3.2), it follows that constructed $\pi_\theta$ is the unique EOT plan between $p_0$ and $p_\theta$: just set $f^*(x_1) \overset{\mathrm{def}}{=} \|x_1\|^2/2 + \epsilon \log v_\theta(x_1)$ in the mentioned theorem. Thus $T_\theta$ is the Schrödinger bridge between $p_0$ and $p_\theta$ by its construction. Then the fact that $T_\theta$ is given by SDE (11) follows from the direct integration of (4) using $\phi_\theta(x_1) \overset{\mathrm{def}}{=} \exp(\frac{\|x_1\|^2}{2\epsilon}) v_\theta(x_1)$ as the Schrödinger potential:

$$g_\theta(x, t) = \epsilon \nabla_x \log \int_{\mathbb{R}^D} \mathcal{N}(x'|x, (1-t)\epsilon I_D) \phi_\theta(x') dx' =$$

$$\epsilon \nabla_x \log \int_{\mathbb{R}^D} \mathcal{N}(x'|x, (1-t)\epsilon I_D) \exp(\frac{\|x'\|^2}{2\epsilon}) v_\theta(x') dx' =$$

$$\epsilon \nabla_x \log \int_{\mathbb{R}^D} \mathcal{N}(x'|x, (1-t)\epsilon I_D) \exp(\frac{\|x'\|^2}{2\epsilon}) \sum_{k=1}^K \alpha_k \mathcal{N}(x'|r_k, \epsilon S_k) dx' =$$

$$\epsilon \nabla_x \log \sum_{k=1}^K \Big\{ \alpha_k \int_{\mathbb{R}^D} \mathcal{N}(x'|x, (1-t)\epsilon I_D) \mathcal{N}(x'|r_k, \epsilon S_k) \exp(\frac{\|x'\|^2}{2\epsilon}) dx' \Big\} =$$

$$\epsilon \nabla_x \log \Big( \underbrace{(2\pi)^{-\frac{D}{2}} |(1-t)\epsilon I_D|^{-\frac{1}{2}}}_{\nabla_x \log \text{ of it} = 0} \sum_{k=1}^K \Big\{ \alpha_k |\epsilon S_k|^{-\frac{1}{2}}$$

$$\int_{\mathbb{R}^D} \exp(-\frac{(x'-x)^T(x'-x)}{2\epsilon(1-t)} - \frac{(x'-r_k) S_k^{-1}(x'-r_k)}{2\epsilon} + \frac{x'^T x'}{2\epsilon}) dx' \Big\} \Big) =$$

$$\epsilon \nabla_x \log \Big( \exp(-\frac{x^T x}{2\epsilon(1-t)}) \sum_{k=1}^K \Big\{ \alpha_k |\epsilon S_k|^{-\frac{1}{2}} \exp(-\frac{r_k^T S_k^{-1} r_k}{2\epsilon})$$

$$\int_{\mathbb{R}^D} \exp(-\frac{1}{2}[x'^T \underbrace{(\frac{t}{\epsilon(1-t)} I_D + \frac{S_k^{-1}}{\epsilon})}_{\overset{\mathrm{def}}{=} A_k^t} x'] + \underbrace{[\frac{1}{\epsilon(1-t)} x + \frac{1}{\epsilon} S_k^{-1} r_k]^T}_{\overset{\mathrm{def}}{=} h_k(x,t)} x') dx' \Big\} \Big) = \quad (20)$$

$$\epsilon \nabla_x \log \Big( \exp(-\frac{x^T x}{2\epsilon(1-t)}) \sum_{k=1}^K \Big\{ \alpha_k |\epsilon S_k|^{-\frac{1}{2}} \exp(-\frac{r_k^T S_k^{-1} r_k}{2\epsilon})$$

$$|A_k^t|^{-\frac{1}{2}} (2\pi)^{\frac{D}{2}} \exp(\frac{1}{2} h_k^T(x,t)(A_k^t)^{-1} h_k(x,t)) \Big\} \Big) = \quad (21)$$

$$\epsilon\nabla_x \log\Big(\underbrace{(2\pi)^{-\frac{D}{2}}\exp(-\frac{x^T x}{2\epsilon(1-t)})}_{\mathcal{N}(x|0,\epsilon(1-t)I_D)}\sum_{k=1}^{K}\Big\{\alpha_k\underbrace{(2\pi)^{-\frac{D}{2}}|\epsilon S_k|^{-\frac{1}{2}}\exp(-\frac{r_k^T S_k^{-1} r_k}{2\epsilon})}_{\mathcal{N}(r_k|0,\epsilon S_k)}$$

$$\underbrace{(2\pi)^{-\frac{D}{2}}|A_k^t|^{-\frac{1}{2}}\exp(\frac{1}{2}h_k^T(x,t)(A_k^t)^{-1}h_k(x,t))}_{\mathcal{N}(h(x,t)|0,A_k^t)}\Big\}\Big) = \qquad(22)$$

$$\epsilon\nabla_x \log\Big(\mathcal{N}(x|0,\epsilon(1-t)I_D)\sum_{k=1}^{K}\big\{\alpha_k\mathcal{N}(r_k|0,\epsilon S_k)\mathcal{N}(h(x,t)|0,A_k^t)\big\}\Big)$$

In the transition from (20) to (21) we use the integral formula from (Petersen et al., 2008, Sec 8.1.1). In the transition from (21) to (22), we simply multiply the expression under $\nabla_x\log$ by $(2\pi)^{-2D}$, as this does not change the expression.

Finally, with the measure disintegration theorem (Vargas et al., 2021, Appendix C), we obtain

$$\mathrm{KL}\left(T^*\|T_\theta\right) = \mathrm{KL}\left(\pi^*\|\pi_\theta\right) + \int_{\mathbb{R}^D\times\mathbb{R}^D}\cancel{\mathrm{KL}\left(T^*_{|x_0,x_1}\|T_{\theta|x_0,x_1}\right)}\pi^\theta(x_0,x_1)dx_0 dx_1 = \mathrm{KL}\left(\pi^*\|\pi_\theta\right).$$

where we cancel out the KL term as it coincides with $\mathrm{KL}\left(W^\epsilon_{|x_0,x_1}\|W^\epsilon_{|x_0,x_1}\right)\equiv 0$. $\qquad\square$

*Proof of Theorem 3.4.* It is intuitively clear that if we are able to approximate $v^*$ arbitrarily well (in some sense) via $v_\theta$, then we also achieve small $\mathrm{KL}\left(\pi^*\|\pi_\theta\right)$ as $\pi_\theta$ explicitly depends on $v_\theta$. The challenge here is that $v^*$ is just a measurable function without any prior known properties, e.g., continuity. Hence, approximating it with a continuous mixture in some reasonable norm, e.g., the uniform norm $\|\cdot\|_\infty$, may be even impossible. This emphasizes the challenge of deriving the desired universal approximation result and points to necessity to use more tricky strategies.

Recall that for all $\delta > 0$ we need to find an unnormalized Gaussian mixture $v_\theta = v_{\theta(\delta)}$ such that $\mathrm{KL}\left(\pi^*\|\pi_\theta\right) < \delta$. To begin with, pick any such $\delta > 0$ and fix it until the end of the proof.

**Stage 1**. This stage is about employing certain known facts from the EOT duality. Let us use

$$\mathrm{Cost}(\pi^*) \stackrel{\text{def}}{=} \int_{\mathbb{R}^D\times\mathbb{R}^D} 1/2\|x_0 - x_1\|^2 d\pi^*(x_0,x_1) + \epsilon\mathrm{KL}\left(\pi^*\|p_0\times p_1\right) \qquad(23)$$

to denote the optimal value of (1). We start from considering the equivalent reformulation of (1):

$$\mathrm{Cost}(\pi^*) =$$
$$\min_{\pi\in\Pi(p_0,p_1)}\left\{\int_{\mathbb{R}^D}\int_{\mathbb{R}^D}\frac{1}{2}\|x_0 - x_1\|^2\pi(x_0,x_1)dx_0 dx_1 + \epsilon\mathrm{KL}\left(\pi\|p_0\times p_1\right)\right\} =$$
$$\epsilon H(p_1) + \underbrace{\min_{\pi\in\Pi(p_0,p_1)}\int_{\mathbb{R}^D}\Big\{\int_{\mathbb{R}^D}\frac{1}{2}\|x_0 - x_1\|^2\pi(x_1|x_0)dx_1 - \epsilon H\big(\pi(\cdot|x_0)\big)\Big\}p_0(x_0)dx_0}_{\stackrel{\text{def}}{=}J^*}, \qquad(24)$$

where $\pi(\cdot|x_0)$ denotes conditional distribution of $x_1$ given $x_0$. Term $J^*$ in (24) is known as the weak representation of EOT, see (Gushchin et al., 2023b, Eq. (3) and (5)) for an extra discussion, and admits a dual form (Backhoff-Veraguas et al., 2019, Eq. (1.3)):

$$J^* = \sup_{f\in\mathcal{C}_{2,b}(\mathbb{R}^D)} J(f) \stackrel{\text{def}}{=} \sup_{f\in\mathcal{C}_{2,b}(\mathbb{R}^D)}\Big\{\int_{\mathbb{R}^D} f^C(x_0)p_0(x_0)dx_0 + \int_{\mathbb{R}^D} f(x_1)p_1(x_1)dx_1\Big\}, \qquad(25)$$

where $\mathcal{C}_{2,b}(\mathbb{R}^D) \stackrel{\text{def}}{=} \{f : \mathbb{R}^D \to \mathbb{R} \text{ continuous s.t. } \exists\alpha,\beta,\gamma\in\mathbb{R}: \alpha\|\cdot\|^2 + \beta \le f(\cdot) \le \gamma\}$ and $f^C$ is the so-called *weak (entropic) C-transform* of $f$ which is defined by

$$f^C(x_0) \stackrel{\text{def}}{=} \inf_{q\in\mathcal{P}_2(\mathbb{R}^D)}\Big\{\int_{\mathbb{R}^D}\frac{1}{2}\|x_0 - x_1\|^2 q(x_1)dx_1 - \epsilon H(q) - \int_{\mathcal{Y}} f(x_1)q(x_1)dx_1\Big\}. \qquad(26)$$

Here $q \in \mathcal{P}_2(\mathbb{R}^D)$ are all the probability distributions whose second moment is finite. We slightly abuse the notation as we write $q(x_1)$ although $q$ here is not necessarily absolutely continuous. However, if $q$ does not have density, then $-\epsilon H(q) = +\infty$, which is a bad option for the minimization problem (26). Therefore, one may consider $q \in \mathcal{P}_{2,ac}(\mathbb{R}^D) \subset \mathcal{P}_2(\mathbb{R}^D)$ in (26). The advantage of EOT compared to many other OT formulations is that the minimizer of (26) is available explicitly:

$$q_{x_0}^f(x_1) \stackrel{\text{def}}{=} \frac{1}{Z^f(x_0)} \exp\left(\frac{f(x_1) - 1/2\|x_0 - x_1\|^2}{\epsilon}\right), \qquad (27)$$

see (Mokrov et al., 2024, Proof of Theorem 1). The mentioned paper considers the compact subsets of $\mathbb{R}^D$ but their derivation is generic and works for our non-compact case as well. Here

$$Z^f(x_0) \stackrel{\text{def}}{=} \int_{\mathbb{R}^D} \exp\left(\frac{f(x_1) - \frac{1}{2}\|x_0 - x_1\|^2}{\epsilon}\right) dx_1 \qquad (28)$$

is the normalizing constant. It is finite thanks to the upper boundness of $f$ due to belonging to $\mathcal{C}_{2,b}(\mathbb{R}^D)$. Due to the same reason, it is not hard to check that $q_x^f$ has a finite second moment. If we further follow the mentioned work and plug $q_{x_0}^f$ in (26), we get $f^C(x_0) = -\epsilon \log Z^f(x_0)$.

From (25) and the definition of the supremum, it follows that for all $\delta' > 0$ there exists some function $\widehat{f} \in \mathcal{C}_{2,b}(\mathbb{R}^D)$ for which the following inequality holds:

$$J(\widehat{f}) = -\epsilon \int_{\mathbb{R}^D} \log Z^{\widehat{f}}(x_0) p_0(x_0) dx_0 + \int_{\mathbb{R}^D} \widehat{f}(x_1) p_1(x_1) dx_1 > \underbrace{\text{Cost}(\pi^*) - \epsilon H(p_1)}_{=J^*} - \delta'.$$

For our needs, we pick $\delta' \stackrel{\text{def}}{=} \frac{\delta\epsilon}{2}$ and suitable $\widehat{f}$ for it and move on to the next stage.

**Stage 2.** Let $\widehat{\gamma} \in \mathbb{R}$ be an upper bound for $\widehat{f}$, i.e., $\widehat{f}(x_1) \leq \widehat{\gamma}$ for all $x_1 \in \mathbb{R}^D$. It exists thanks to $\widehat{f} \in \mathcal{C}_{2,b}(\mathbb{R}^D)$. Recall that $p_1$ is compactly supported by the assumption of the theorem. Let $R > 0$ be some radius such that the zero-centered ball of this radius contains the support of $p_1$. We define

$$\widetilde{f}(x_1) \stackrel{\text{def}}{=} \widehat{f}(x_1) - \max\{0, \|x_1\|^2 - R^2\} \leq \widehat{f}(x_1) \leq \widehat{\gamma}.$$

We see that

$$\widetilde{f} \leq \widehat{f} \implies Z^{\widetilde{f}} \leq Z^{\widehat{f}} \implies \widetilde{f}^C \geq \widehat{f}^C \implies \int_{\mathbb{R}^D} \widetilde{f}^C(x_0) p_0(x_0) dx_0 \geq \int_{\mathbb{R}^D} \widehat{f}^C(x_0) p_0(x_0) dx_0. \quad (29)$$

By the construction of $\widetilde{f}$, it holds that $\widetilde{f}(x_1) = \widehat{f}(x_1)$ when $x_1$ is in the support of $p_1$. Thus,

$$\int_{\mathbb{R}^D} \widetilde{f}(x_1) p_1(x_1) dx_1 = \int_{\mathbb{R}^D} \widehat{f}(x_1) p_1(x_1) dx_1. \qquad (30)$$

We combine (29) with (30) and see that $J(\widetilde{f}) \geq J(\widehat{f}) > J^* - \delta' = \text{Cost}(\pi^*) - \epsilon H(p_1) - \delta'$.

We note that $p_0$ is compactly supported, and $Z^{\widetilde{f}}$ is continuous (w.r.t. $x_0$) and non-negative. Therefore, there exists a constant $z_{\min} > 0$ such that $Z^{\widetilde{f}}(x_0) \geq z_{\min}$ when $x_0$ belongs to the support of $p_0$. Analogously, since $p_1$ is compactly supported, we may find a positive constant $e_{\min} > 0$ such that $\frac{1}{2}\exp(\widetilde{f}(x_1)/\epsilon) \geq e_{\min}$ for all $x_1$ in the support of $p_1$. We fix constants $z_{\min}, e_{\min}$ for next steps.

Right now we derive

$$\exp\left(\widetilde{f}(x_1)/\epsilon\right) \leq \exp\left(\frac{\widehat{\gamma} - \max\{0, \|x_1\|^2 - R^2\}}{\epsilon}\right) \leq \exp\left(\frac{\widehat{\gamma} + R^2}{\epsilon}\right) \cdot \exp(-\|x_1\|^2/\epsilon).$$

This means that $x_1 \mapsto \exp\left(\widetilde{f}(x_1)/\epsilon\right)$ is a normalizable density ($\int_{\mathbb{R}^D} \exp\left(\widetilde{f}(x_1)/\epsilon\right) dx_1 < \infty$) because its density is bounded by an unnormalized Gaussian density. Additionally, we see that $\exp\left(\widetilde{f}(x_1)/\epsilon\right)$ vanishes at infinity. Thus, for every $\delta'' > 0$ there exists an unnormalized[2] Gaussian mixture $v_{\widetilde{\theta}} = v_{\widetilde{\theta}(\delta'')}$ (Nguyen et al., 2020, Theorem 5a) which is $\delta''$-close to $\exp(\widetilde{f}/\epsilon)$ in the uniform norm on $\mathbb{R}^D$:

$$\|v_{\widetilde{\theta}} - \exp(\widetilde{f}/\epsilon)\|_\infty = \sup_{x_1 \in \mathbb{R}^D} |v_{\widetilde{\theta}}(x_1) - \exp(\widetilde{f}(x_1)/\epsilon)| < \delta''. \qquad (31)$$

---

[2]The result of (Nguyen et al., 2020) considers the approximation of *normalized* mixtures. This detail is not important and the result straightforwardly extends to *unnormalized* mixtures. One can first normalize the mixture, approximate it and then re-scale both the target and the approximator back to the original scale.

From the statement of the mentioned theorem it also follows that one may pick all the covariances in the mixture $v_{\widetilde{\theta}}$ to be **scalar**, i.e., $v_{\widetilde{\theta}}(x_1) = \sum_{k=1}^{K} \beta_k \mathcal{N}(x_1 | \mu_k, \epsilon \sigma_k^2 I)$ for some $K$ and $\mu_k \in \mathbb{R}^D$, $\sigma_k \in \mathbb{R}_+$ ($k \in \{1, \dots, K\}$). Indeed, just recall the definition of $\mathcal{M}_m^g$ in (Nguyen et al., 2020) and put $g$ to be a standard $D$-dimensional normal distribution. For further derivations, we pick

$$\delta'' \stackrel{\text{def}}{=} \min \left\{ \delta/2 \left( \frac{1}{e_{\min}} + \frac{(2\pi\epsilon)^{D/2}}{z_{\min}} \right)^{-1}, e_{\min} \right\}, \tag{32}$$

and its respective mixture $v_{\widetilde{\theta}}$ with scalar components' covariances. We define $v_\theta(x_1) \stackrel{\text{def}}{=} v_{\widetilde{\theta}}(x_1) \exp(-\frac{\|x_1\|^2}{2\epsilon})$. It is again an unnormalized Gaussian mixture because it is a product of two unnomalized Gaussian mixtures. Besides, it also has scalar covariances of its components because multiplier $\exp(-\frac{\|x_1\|^2}{2\epsilon})$'s covariance is scalar itself. More precisely, we have

$$v_\theta(x_1) \stackrel{\text{def}}{=} v_{\widetilde{\theta}}(x_1) \exp(-\frac{\|x_1\|^2}{2\epsilon}) = \sum_{k=1}^{K} \beta_k \mathcal{N}(x_1 | \mu_k, \epsilon \sigma_k^2 I) \exp(-\frac{\|x_1\|^2}{2\epsilon}) =$$

$$(\sqrt{2\pi\epsilon})^D \sum_{k=1}^{K} \beta_k \mathcal{N}(x_1 | \mu_k, \epsilon \sigma_k^2 I) \mathcal{N}(x_1 | 0, \epsilon I) =$$

$$\sum_{k=1}^{K} \underbrace{(\sqrt{2\pi\epsilon})^D \beta_k \mathcal{N}(0 | \mu_k, \epsilon(1+\sigma_k^2) I)}_{\stackrel{\text{def}}{=} \alpha_k} \mathcal{N}(x_1 | \underbrace{\frac{\mu_k}{1+\sigma_k^2}}_{\stackrel{\text{def}}{=} r_k}, \epsilon \underbrace{\frac{\sigma_k^2}{\sigma_k^2+1}}_{\stackrel{\text{def}}{=} \lambda_k} I) = \sum_{k=1}^{K} \alpha_k \mathcal{N}(x_1 | r_k, \epsilon \lambda_k I). \tag{33}$$

Here in transition to (33) we use the formulas from (Petersen et al., 2008, §8.1.8). We derive that

$$Z^{\widetilde{f}}(x_0) = \int_{\mathbb{R}^D} \exp\left(\widetilde{f}(x_1)/\epsilon\right) \exp\left(\frac{-1/2\|x_0 - x_1\|^2}{\epsilon}\right) dx_1 >$$

$$\int_{\mathbb{R}^D} \left(v_{\widetilde{\theta}}(x_1) - \delta''\right) \exp\left(\frac{-1/2\|x_0 - x_1\|^2}{\epsilon}\right) dx_1 =$$

$$\int_{\mathbb{R}^D} v_{\widetilde{\theta}}(x_1) \exp\left(\frac{-1/2\|x_0 - x_1\|^2}{\epsilon}\right) dx_1 - \delta'' \underbrace{\int_{\mathbb{R}^D} \exp\left(\frac{-1/2\|x_0 - x_1\|^2}{\epsilon}\right) dx_1}_{=(2\pi\epsilon)^{D/2}} =$$

$$\exp(-\frac{\|x_0\|^2}{2\epsilon}) \int_{\mathbb{R}^D} v_\theta(x_1) \exp\left(\langle x_0, x_1 \rangle/\epsilon\right) dx_1 - \delta''(2\pi\epsilon)^{D/2},$$

or, equivalently,

$$Z^{\widetilde{f}}(x_0) + \delta''(2\pi\epsilon)^{D/2} > \exp(-\frac{\|x_0\|^2}{2\epsilon}) \underbrace{\int_{\mathbb{R}^D} v_\theta(x_1) \exp\left(\langle x_0, x_1 \rangle/\epsilon\right) dx_1}_{=c_\theta(x_0)} = \exp(-\frac{\|x_0\|^2}{2\epsilon}) c_\theta(x_0).$$

$$\tag{34}$$

Since $z \mapsto \log z$ is a $\frac{1}{z_{\min}}$-Lipschitz function on $[z_{\min}, +\infty)$ and $Z^{\widetilde{f}}(x_0) \geq z_{\min}$ for all $x_0$ in the support of $p_0$, we may write

$$\frac{\delta''(2\pi\epsilon)^{D/2}}{z_{\min}} \geq \log\left(Z^{\widetilde{f}}(x_0) + \delta''(2\pi\epsilon)^{D/2}\right) - \log\left(Z^{\widetilde{f}}(x_0)\right).$$

We use this inequality with (34) to derive

$$\log\left(Z^{\widetilde{f}}(x_0)\right) + \frac{\delta''(2\pi\epsilon)^{D/2}}{z_{\min}} \geq \log\left(Z^{\widetilde{f}}(x_0) + \delta''(2\pi\epsilon)^{D/2}\right) > -\frac{\|x_0\|^2}{2\epsilon} + \log c_\theta(x)$$

for all $x_0$ in the support of $p_0$. We integrate this expression for $x_0 \sim p_0$ and obtain

$$\int_{\mathbb{R}^D} \log Z^{\widetilde{f}}(x_0) p_0(x_0) dx_0 + \frac{\delta''(2\pi\epsilon)^{D/2}}{z_{\min}} > -\int_{\mathbb{R}^D} \frac{\|x_0\|^2}{2\epsilon} p_0(x_0) dx_0 + \int_{\mathbb{R}^D} \log c_\theta(x_0) p_0(x_0) dx_0.$$

After regrouping the terms, we get

$$\int_{\mathbb{R}^D} \frac{\|x_0\|^2}{2\epsilon} p_0(x_0)dx_0 - \int_{\mathbb{R}^D} \log c_\theta(x_0)p_0(x_0)dx_0 + \frac{\delta''(2\pi\epsilon)^{D/2}}{z_{\min}} > - \int_{\mathbb{R}^D} \log Z^{\widetilde{f}}(x_0)p_0(x_0)dx_0. \tag{35}$$

Now we study another expression. Recalling that $z \mapsto \log(z)$ is $\frac{1}{e_{\min}}$-Lipschitz for $z \geq e_{\min}$, we get

$$\frac{\delta''}{e_{\min}} \geq \log \exp(\widetilde{f}(x_1)/\epsilon) - \log\big[\exp(\widetilde{f}(x_1)/\epsilon) - \delta''\big] = \widetilde{f}(x_1)/\epsilon - \log\big[\exp(\widetilde{f}(x_1)/\epsilon) - \delta''\big] \tag{36}$$

for all $x_1$ in the support of $p_1$. Here we also use the fact that

$$\exp(\widetilde{f}(x_1)/\epsilon) - \delta'' \geq \exp(\widetilde{f}(x_1)/\epsilon) - e_{\min} \geq 2e_{\min} - e_{\min} \geq e_{\min}$$

by the choice of $\delta''$, recall the definition of $e_{\min}$ and see (32). We recall (31) to get

$$\log v_{\widetilde{\theta}}(x_1) \overset{(31)}{>} \log\big[\exp(\widetilde{f}(x_1)/\epsilon) - \delta''\big] \overset{(36)}{\geq} \widetilde{f}(x_1)/\epsilon - \frac{\delta''}{e_{\min}}.$$

We exploit this observation to derive

$$\int_{\mathbb{R}^D} \frac{\|x_1\|^2}{2\epsilon} p_1(x_1)dx_1 + \int_{\mathbb{R}^D} \log v_\theta(x_1)p_1(x_1)dx_1 = \int_{\mathbb{R}^D} \log v_{\widetilde{\theta}}(x_1)p_1(x_1)dx_1 >$$

$$\int_{\mathbb{R}^D} \big(\frac{\widetilde{f}(x_1)}{\epsilon} - \frac{\delta''}{e_{\min}}\big)p_1(x_1)dx_1 = \int_{\mathbb{R}^D} \frac{\widetilde{f}(x_1)}{\epsilon} p_1(x_1)dx_1 - \frac{\delta''}{e_{\min}}. \tag{37}$$

We sum (37) with (35) and get

$$\overbrace{\int_{\mathbb{R}^D} \log v_\theta(x_1)p_1(x_1)dx_1 - \int_{\mathbb{R}^D} \log c_\theta(x)p_0(x_0)dx_0}^{=-\mathcal{L}(\theta)} >$$

$$\underbrace{- \int_{\mathbb{R}^D} \log Z^{\widetilde{f}}(x_0)p_0(x_0)dx_0 + \int_{\mathbb{R}^D} \frac{\widetilde{f}(x_1)}{\epsilon} p_1(x_1)dx_1 - \frac{\delta''}{e_{\min}} - \frac{\delta''(2\pi\epsilon)^{D/2}}{z_{\min}} -}_{=\epsilon^{-1}J(\widetilde{f})}$$

$$\int_{\mathbb{R}^D} \frac{\|x_0\|^2}{2\epsilon} p_0(x_0)dx_0 - \int_{\mathbb{R}^D} \frac{\|x_1\|^2}{2\epsilon} p_1(x_1)dx_1 =$$

$$\epsilon^{-1}J(\widetilde{f}) - \frac{\delta''}{e_{\min}} - \frac{\delta''(2\pi\epsilon)^{D/2}}{z_{\min}} - \int_{\mathbb{R}^D} \frac{\|x_0\|^2}{2\epsilon} p_0(x_0)dx_0 - \int_{\mathbb{R}^D} \frac{\|x_1\|^2}{2\epsilon} p_1(x_1)dx_1 >$$

$$\epsilon^{-1}\big[\text{Cost}(\pi^*) - \epsilon H(p_1) - \frac{\delta\epsilon}{2}\big] - \frac{\delta''}{e_{\min}} - \frac{\delta''(2\pi\epsilon)^{D/2}}{z_{\min}} - \int_{\mathbb{R}^D} \frac{\|x_0\|^2}{2\epsilon} p_0(x_0)dx_0 - \int_{\mathbb{R}^D} \frac{\|x_1\|^2}{2\epsilon} p_1(x_1)dx_1 =$$

$$\underbrace{\epsilon^{-1}\bigg[\text{Cost}(\pi^*) - \epsilon H(p_1) - \int_{\mathbb{R}^D} \frac{\|x_0\|^2}{2} p_0(x_0)dx_0 - \int_{\mathbb{R}^D} \frac{\|x_1\|^2}{2} p_1(x_1)dx_1\bigg]}_{=-\mathcal{L}^* \text{ in (18)}} - \frac{\delta}{2} - \frac{\delta''}{e_{\min}} - \frac{\delta''(2\pi\epsilon)^{D/2}}{z_{\min}} =$$

$$-\mathcal{L}^* - \frac{\delta}{2} - \frac{\delta''}{e_{\min}} - \frac{\delta''(2\pi\epsilon)^{D/2}}{z_{\min}}, \tag{38}$$

where $\mathcal{L}^*$ matches the constant defined in (18). Indeed,

$$-\mathcal{L}^* = \underbrace{-H(\pi^*) + H(p_0)}_{=\text{KL}(\pi^*\|p_0\times p_1) - H(p_1)} -\epsilon^{-1} \int_{\mathbb{R}^D\times\mathbb{R}^D} \langle x_0, x_1\rangle \pi^*(x_0, x_1)dx_0dx_1 =$$

$$-H(p_1) + \text{KL}(\pi^*\|p_0 \times p_1) \underbrace{- \epsilon^{-1} \int_{\mathbb{R}^D\times\mathbb{R}^D} \langle x_0, x_1\rangle \pi^*(x_0, x_1)dx_0dx_1}_{=\epsilon^{-1}\big(\text{Cost}(\pi^*)+1/2\int_{\mathbb{R}^D}\|x_0\|^2p_0(x_0)dx_0+1/2\int_{\mathbb{R}^D}\|x_1\|^2p_1(x_1)dx_1\big)} =$$

$$\epsilon^{-1}\bigg[\text{Cost}(\pi^*) - \epsilon H(p_1) - \int_{\mathbb{R}^D} \frac{\|x_0\|^2}{2} p_0(x_0)dx_0 - \int_{\mathbb{R}^D} \frac{\|x_1\|^2}{2} p_1(x_1)dx_1\bigg].$$

At the same time, from (19) and (38) we derive that

$$\text{KL}\left(\pi^*\|\pi_\theta\right) = \mathcal{L}(\theta) - \mathcal{L}^* < \frac{\delta}{2} + \delta''\left\{\frac{1}{e_{\min}} + \frac{(2\pi\epsilon)^{D/2}}{z_{\min}}\right\} \leq \frac{\delta}{2} + \frac{\delta}{2} = \delta.$$

Thus, Gaussian mixture $v_\theta$ is a one that we seek for. This finishes the proof. $\qquad\square$

### B.2 PROOF OF THEOREM A.1 IN APPENDIX A

The proof follows from combining the two auxiliary facts below.

**Proposition B.1** (Rademacher bound on the statistical error). *It holds that*

$$\mathbb{E}\left[\mathcal{L}(\widehat{\theta}) - \mathcal{L}(\theta^*)\right] \leq 4\mathcal{R}_N(\mathcal{V}_0, p_0) + 4\mathcal{R}_M(\mathcal{V}_1, p_1),$$

*where $\mathcal{V}_0 = \{\log c_\theta | \theta \in \Theta\}$, $\mathcal{V}_1 = \{\log v_\theta | \theta \in \Theta\}$ and $\mathcal{R}_N(\mathcal{V}, p)$ denotes the well-celebrated Rademacher complexity (Shalev-Shwartz & Ben-David, 2014, §26) of the functional class $\mathcal{V}$ w.r.t. to the sample size $N$ of distribution $p$.*

*Proof of Proposition B.1.* This result can be derived exactly the same way as (Mokrov et al., 2024, Theorem 4) or (Taghvaei & Jalali, 2019, Theorem 3.4). $\qquad\square$

**Proposition B.2** (Rademacher complexity bound for constrained log-sum-exp quadratic functions). *Let $0 < a \leq A$, let $0 < u \leq U$, let $0 < w \leq W$ and $V > 0$. Consider the class of functions*

$$\mathcal{V} = \Big\{x \mapsto \log \sum_{k=1}^K \alpha_k \exp\left(x^T U_k x + v_k^T x + w_k\right) \text{ with} \tag{39}$$

$$uI \preceq U_k = U_k^T \preceq UI; \|v_k\| \leq V; w \leq w_k \leq W; a \leq \alpha_k \leq A\Big\}.$$

*We say that such class is the class of **constrained** log-sum-exp quadratic functions. Assume that $p$ is compactly supported and the support lies in a zero-centered ball of a radius $P > 0$. Then*

$$\mathcal{R}_N(\mathcal{V}, p) \leq \frac{C}{\sqrt{N}},$$

*where the constant $C$ depends only on $K, u, U, a, A, V, w, W, P$ but not on the sample size $N$.*

*Proof of Proposition B.2.* The Rademacher complexity of linear constrained functions $x \mapsto v_k^T x + w_k$ is well known and is bounded by $O(\frac{1}{\sqrt{N}})$, see (Shalev-Shwartz & Ben-David, 2014, §26.2). The complexity of the constrained quadratic functions $x \mapsto x^T U_k x$ is also $O(\frac{1}{\sqrt{N}})$ which follows from their representation using the Reproducing Kernel Hilbert spaces (RKHS), see (Latorre et al., 2021, Lemma 5 & Eq. 24) and additionally (Mohri et al., 2018, Theorem 6.12). Hence, by the well-known additivity of the Rademacher complexity it also follows that the complexity of constrained forms $x \mapsto x^T U_k x + v_k^T x + w_k$ is also bounded by $O(\frac{1}{\sqrt{N}})$. Since $x, U_k, v_k, w_k$ are bounded, the function $x \mapsto \exp\left(x^T U_k x + v_k^T x + w_k\right)$ is Lipschitz in $x$ with the shared Lipschitz constant for all admissible $U_k, v_k, w_k$. Therefore, the Rademacher complexity of such functions is also $O(\frac{1}{\sqrt{N}})$, recall the Talagrand's contraction principle (Mohri et al., 2018, Lemma 5.7). The same applies to

$$x \mapsto \alpha_k \exp\left(x^T U_k x + v_k^T x + w_k\right) = \exp\left(x^T U_k x + v_k^T x + [w_k + \log \alpha_k]\right)$$

as these are also constrained exp-quadratic forms but with slightly adjusted constraints on the bias parameter $w_k$. Using the additivity of the Rademacher complexity again, we see that $K$-sums

$$x \mapsto \sum_{k=1}^K \alpha_k \exp\left(x^T U_k x + v_k^T x + w_k\right)$$

of such functions also have complexity bounded by $O(\frac{1}{\sqrt{N}})$. The remaining step is to note that each such function is both lower and upper bounded (by some *positive numbers* depending on the constraints), hence the logarithm of such functions is also a Lipschitz operation with some finite Lipschitz constant. Thus, the complexity of $x \mapsto \log \sum_{k=1}^K \alpha_k \exp\left(x^T U_k x + v_k^T x + w_k\right)$ is also $O(\frac{1}{\sqrt{N}})$; the constant hidden in $O(\cdot)$ incapsulates the dependedce on $K, u, U, a, A, V, w, W, P$. $\qquad\square$

Finally, we can prove the bound on the estimation error in Theorem A.1.

*Proof of Theorem A.1.* Just note that both $\mathcal{V}_0$ and $\mathcal{V}_1$ are the constrained classes of log-sum-exp quadratic functions in the sense of Proposition B.2 and apply Proposition B.1. For $\mathcal{V}_1$ this directly follows from the assumptions of the current Theorem. For $\mathcal{V}_0$ it follows from the the fact that $c_\theta$ is also a log-sum-exp quadratic function with constrained parameters (our Proposition 3.2). $\qquad\square$

## C   EMBRYONIC STEM CELL DIFFERENTIATION SINGLE CELL DATA

For the embryonic stem cell differentiation single cell setup we use code and data from

<p align="center">https://github.com/KrishnaswamyLab/TrajectoryNet</p>

to work with the embryonic stem cell differentiation dataset and to evaluate our light solver.

The provided data shows the cell differentiation collected at five different intervals ($t_0$: day 0 to 3, $t_1$: day 6 to 9, $t_2$: day 12 to 15, $t_3$: day 18 to 21, $t_4$: day 24 to 27). These collected cells were analysed by scRNAseq subjected to quality control filtering and then represented as feature vectors using Principal Component Analysis (PCA).

| Solver | $\mathbb{W}_1$ metric |
|---|---|
| OT-CFM | 0.79 $\pm$0.068 |
| [SF]$^2$M-Exact | 0.793 $\pm$0.066 |
| **LightSB (ours)** | 0.823 $\pm$0.017 |
| Reg. CNF | 0.825 $\pm$ |
| T. Net | 0.848 $\pm$ |
| DSB | 0.862 $\pm$0.023 |
| I-CFM | 0.872 $\pm$0.087 |
| [SF]$^2$M-Geo | 0.879 $\pm$0.148 |
| [SF]$^2$M-Sink | 1.198 $\pm$0.342 |
| SB-CFM | 1.221 $\pm$0.380 |
| DSBM | 1.775 $\pm$0.429 |

Table 4: The quality of intermediate distribution restoration of single-cell data by different methods.

Following the above-mentioned works, we consider solving the problem of transporting the cell distribution at time $t_{i-1}$ to time $t_{i+1}$ for $i \in [1, 2, 3]$. Then we predict the cell distributions at the intermediate time $t_i$ and compute the Wasserstein-1 ($\mathbb{W}_1$) distance between the predicted distribution and the ground truth distribution. We average over all 3 setups $i \in [1, 2, 3]$ and present our results in Table 4. To estimate the standard deviation, we run 5 experiments with different seeds for each $i \in [1, 2, 3]$. For LightSB we use $K = 100$, lr $= 10^{-2}$, $\epsilon = 0.1$, batch size 128 and do $2 \cdot 10^3$ gradient steps. We use results for other methods from (Tong et al., 2023), whose authors were the first to consider this setup in (Tong et al., 2020). Our solver ($\epsilon$=0.1) performs at the level of the best other methods, *while converging only in 1 minute on 4 CPU cores and learning only $\sim$1000 parameters.*

## D   DETAILS OF THE EXPERIMENTS

### D.1   GENERAL IMPLEMENTATION DETAILS

To minimize (10), we parameterize $\alpha_k$, $r_k$ and $S_k$ of $v_\theta$ (9) and use the Adam optimiser (Kingma & Ba, 2014). We parameterize $\alpha_k$ as using the logarithm $\log \alpha_k$; we parameterize $r_k$ directly as a vector; we parameterise the matrix $S_k$ in the diagonal form with the values $\log(S_k)_{i,i}$ on its diagonal.

**Initialization.** We initialize $\log \alpha_k$ by $\log \frac{1}{K}$, $r_k$ by using random samples from $p_1$ and $\log(S_k)_{i,i}$ by $\log 0.1$ (it is can be tuned as a hyperparameter but even without any tuning it works well with this initialisation on every considered experimental setup).

### D.2   DETAILS OF TOY EXPERIMENTS

We use $K = 500$ in all the cases. For $\epsilon = 10^{-1}$ and $\epsilon = 10^{-2}$, we use $lr = 10^{-3}$ and for $\epsilon = 2 \cdot 10^{-3}$ we use $lr = 10$ and batchsize 128. We do $10^4$ gradient steps.

### D.3   DETAILS OF EVALUATION ON THE BENCHMARK

We use $K = 50$ in all the cases. For $\epsilon = 10^{-1}$, we use $lr = 10^{-3}$. For $\epsilon = 2 \cdot 10^{-3}$, we use Adam with $lr = 10$ and batch size 128. We do $10^4$ gradient steps.

In Table 5, we additionally present results of the non-conditional $\mathrm{B}\mathbb{W}_2^2$-UVP metric. LightSB beats other solvers or performs at the same level for all setups.

|  | $\epsilon = 0.1$ | | | | $\epsilon = 1$ | | | | $\epsilon = 10$ | | | |
|---|---|---|---|---|---|---|---|---|---|---|---|---|
|  | $D=2$ | $D=16$ | $D=64$ | $D=128$ | $D=2$ | $D=16$ | $D=64$ | $D=128$ | $D=2$ | $D=16$ | $D=64$ | $D=128$ |
| Best solver | 0.016 | 0.05 | 0.25 | 0.22 | **0.005** | 0.09 | 0.56 | 0.12 | **0.01** | **0.02** | **0.15** | **0.23** |
| $\lfloor$**LightSB**$\rceil$ | **0.005** | **0.017** | **0.037** | **0.069** | 0.004 | **0.01** | **0.03** | **0.07** | 0.03 | 0.04 | 0.17 | 0.30 |
| **± std** | **±0.002** | **±0.007** | **±0.007** | **±0.008** | **±0.002** | **±0.004** | **±0.006** | **±0.007** | **±0.01** | **±0.01** | **±0.01** | **±0.02** |

Table 5: Comparisons of $\mathbb{BW}_2^2$-UVP $\downarrow$ (%) between the target $\mathbb{P}_1$ and learned right marginal of $\pi_\theta$.

### D.4 DETAILS OF MULTIMODAL SINGLE-CELL INTEGRATION EXPERIMENTS

We use data from the Kaggle competition "Open Problems - Multimodal Single-Cell Integration":

```
https://www.kaggle.com/competitions/open-problems-multimodal
```

The data describes gene expression of cells at days 2, 3, 4 and 7 which containt $6071, 7643, 8485, 7195$ data points, respectively. Analogously to Tong et al. (2023), we use only CITEseq expression data; to remove the donor-dependent bias we select only one donor with ID 13176. To preprocess the data, we use PCA projections with $50, 100, 1000$ components. Then for each case we consider 2 setups: data from day 2 and day 4 as a distribution pair for the SB problem with data from day 3 for evaluation, and data from day 3 and day 7 as a distribution pair for the SB problem with data from day 4 for evaluation. In each setup, we normalize the data by scaling it to the sum of each feature variance over the concatenated data from the start, end, and evaluation days (after PCA projection). For the evaluation, we take the prediction for the evaluation day for all cells from the start day and calculate the energy distance with the ground truth distribution.

For all described setups we use $\epsilon = 0.1$ in our and baseline solvers. For our LightSB solver we use $K = 10$, $lr = 10^{-2}$ and batchsize 128. We do $10^4$ gradient steps.

**Baselines**. We compare LightSB with SB/EOT algorithms from three different classes: maximin (Gushchin et al., 2023a), Langevin-based (Mokrov et al., 2024) and IPF-based (Vargas et al., 2021). For completeness, we also add the popular discrete EOT Sinkhorn solver (Cuturi, 2013).

1. **Maximin solver.** For (Gushchin et al., 2023a) solver we use the official code from

   ```
   https://github.com/ngushchin/EntropicNeuralOptimalTransport
   ```

   We use the same hyperparameters for this setup as the authors (Gushchin et al., 2023a, Appendix E) use in their high-dimensional Gaussian setup. The only exception is the number of discretization steps N, which we set to 100 as well as for SB solver (Vargas et al., 2021) below.

2. **IPF-based solver.** For (Vargas et al., 2021) solver we use the official code from

   ```
   https://github.com/franciscovargas/GP_Sinkhorn
   ```

   Instead of Gaussian processes which the authors use, we use the same neural network as in (Gushchin et al., 2023b) to get better scalability. We use $N = 100$ discretization steps, 50 IPF iterations, 10 epochs on the each IPF-iteration and 128 samples from distributions $p_0$ and $p_1$ in each of them. We use the Adam optimizer with $lr = 10^{-4}$ for optimization.

3. **Langevin-based solver**. For (Mokrov et al., 2024) solver, we use the official code from

   ```
   https://github.com/PetrMokrov/Energy-guided-Entropic-OT
   ```

   We take the advantage of the author's setup from their 2D Gaussian→Swissroll experiment. Following our experimental framework, we adapt the original code by increasing the dimensionality of the learned fully-connected NN potentials. The chosen hidden directionalities for the potentials are `[256, 256, 256]` for $D = 50, 100$ and `[2048, 1024, 512]` for $D = 1000$. We choose all hyperparameters of the method in concordance with their code for $\varepsilon = 0.1$ EOT regularization coefficient, except $lr$. We pick $lr = 5 \cdot 10^{-4}$ for training stability reasons. The numbers of training iterations are $N = 8K$ for $D = 50, 100$ and $N = 4K$ for $D = 1000$. We get predictions for intermediate distributions by using Brownian Bridge (analogous to (14)).

4. **Discrete solver.**[3] To run the Sinkhorn algorithm (Cuturi, 2013), we use the `ot.sinkhorn` with parameters `method="sinkhorn_log"` and `stop_threshold=1e-8` procedure from

---

[3]Discrete OT neither can be straightforwardly used to infer trajectories for new (out-of-train-sample) input cells, nor it provides the trajectories for existing cells. In our setup, the latter issue can be overcome by inserting

Python OT Package (Flamary et al., 2021). Note the default threshold parameter is $10^{-9}$ but we found that the algorithm stucks at tolerance $\approx 10^{-8}$; hence, we increased the tolerance.

*Remark.* We use the full-dataset (a.k.a. full-batch) discrete OT. We found that for $\epsilon = 0.1$ (which we use for all the solvers) it converges very slowly, even requiring more time to converge than our light solver in dimension 1000. This is explainable the convergence of Sinkhorn algorithm notably degrades when $\epsilon \to 0$; it empirically seems like this degradation is worse that in our solver. We demonstrate the convergence plots of the Sinkhorn vs. our light solver in Figure 4.

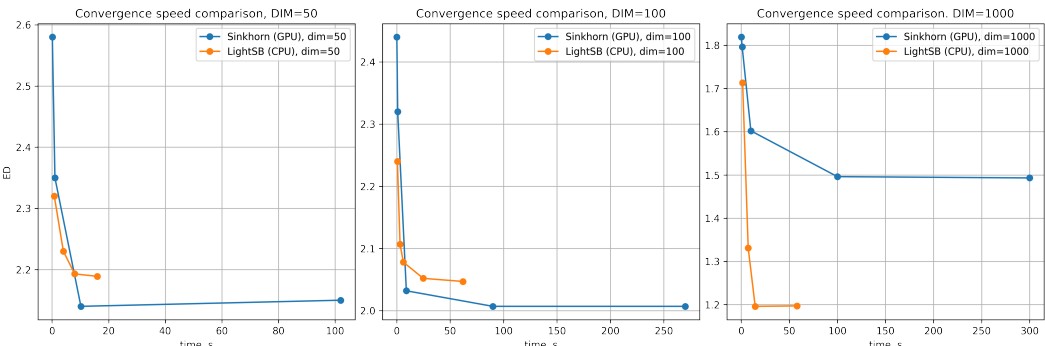

Figure 4: Convergence speed comparison on MSCI dataset, starting day 3, ending day 7 and evaluation day 4.

### D.5 DETAILS OF IMAGE DATA EXPERIMENTS

We use the official ALAE code and model from

$$\texttt{https://github.com/podgorskiy/ALAE}$$

and neural network extracted attributes for the FFHQ dataset from

$$\texttt{https://github.com/DCGM/ffhq-features-dataset}$$

We use $K = 10$, $lr = 10^{-3}$ and batchsize 128. We do $10^4$ gradient steps.

## E COMPLETE PARAMETERIZATION OF THE EOT PLAN

As we pointed in §3.1, our solver obtains an approximate density $\pi_\theta(x_1|x_0)$ of conditional distributions $\pi^*(x_1|x_0)$ but does not recover the density of the entire plan $\pi_\theta(x_0, x_1) \approx \pi^*(x_0, x_1)$. However, in some practical applications it may be needed to recover this density. Fortunately, this requires only a **minor modification** of our light solver. Indeed, consider the parameterization

$$\pi_{\omega,\theta}(x_0, x_1) = p_\omega(x_0)\pi_\theta(x_1|x_0) = p_\omega(x_0)\frac{\exp\left(\langle x_0, x_1\rangle/\epsilon\right)v_\theta(x_1)}{c_\theta(x_0)} \tag{40}$$

which is analogous to (7), but we also introduce a density model $p_\omega$ for the left marginal of the plan. By repeating the derivations of the proof of Proposition 3.1, it is not hard to see that

$$\text{KL}\left(\pi^*\|\pi_{\omega,\theta}\right) = \text{KL}\left(p_0\|p_\omega\right) + \left[\mathcal{L}(\theta) - \mathcal{L}^*\right], \tag{41}$$

which means that learning of parameters $\omega$ turns to be a **separate** density estimation problem. It can be solved, e.g., with the well-celebrated expectation-maximization algorithm (Dempster et al., 1977) for Gaussian mixture models or with a normalizing flow (Rezende & Mohamed, 2015).

## F RELATED WORK: OTHER OT SOLVERS

For completeness of the exposition, we mention OT solvers which are not directly relevant to our EOT/SB setup because of considering non-entropic OT or discrete OT settings.

---

the Brownian Bridge (analogously to (14)) on top of pairs of samples from the recovered discrete EOT plan (Stromme, 2023). Sampling from this bridge, one may construct an approximation of the distribution at the intermediate time of interest.

**Discrete OT solvers.** There exist many OT solvers (Cuturi, 2013; Dvurechensky et al., 2018), (Nguyen et al., 2022; Xie et al., 2022) for the discrete OT setup (Peyré et al., 2019). This setup requires finding a discrete matching between given train samples but does not require generalization on the test data. Hence, discrete solvers are not relevant to our continuous EOT/SB setup (§2). It is worth mentioning that there are several works studying the statistical properties of OT and developing out-of-sample estimators based on the discrete/batched OT solutions (Hütter & Rigollet, 2021; Pooladian & Niles-Weed, 2021; Manole et al., 2021; Deb et al., 2021), Rigollet & Stromme (2022); Fatras et al. (2020). Despite having good theoretical properties, they mostly estimate the barycentric projection but not the entire plan $\pi^*$.

**Other continuous OT solvers.** Above in the work, we discuss the EOT/SB solvers but there exist many papers proposing neural solvers for other OT formulations: unregularized OT (Henry-Labordere, 2019; Makkuva et al., 2020; Korotin et al., 2021a;b; 2022b;a; Fan et al., 2023; Liu et al., 2022; Gazdieva et al., 2023; Rout et al., 2021; Amos, 2022), weak OT (Korotin et al., 2023b;a), unbalanced OT (Choi et al., 2023; Yang & Uhler, 2018) general OT (Asadulaev et al., 2024). These works are of limited relevance as they do not solve EOT/SB.

## G  LIMITATIONS AND BROADER IMPACT

**Limitations**. We summarize and list some limitations of our light solver.

1. Analogously to the well-celebrated Sinkhorn algorithm (Cuturi, 2013) for the discrete OT, our solver may experience computational instabilities when applied to very small regularization coefficients $\epsilon > 0$. This is due to the necessity to compute the exponent of values which proportional to $\epsilon^{-1}$ when computing $c_\theta$ or $v_\theta$ in (8). However, as our experiments show (§5), our light solver actually works well for reasonably small $\epsilon$.

2. Our main optimization objective (8) is not necessarily convex w.r.t. the parameters $\theta$, i.e., the gradient-based optimization methods may experience local minima. While we mention this issue, we do not consider it serious enough: anyway, many existing machine learning algorithms have non-convex objectives ($k$-means, Gaussian mixture models, deep neural networks, etc.) and do not necessarily converge to the global optimum but still work well in downstream tasks. Note that the existing alternative continuous EOT/SB solvers (§4) also have non-convex objectives.

3. Our solver uses a kind of a Gaussian mixture approximation (9) of EOT/SB which may be too restrictive to apply our solver to large-scale generative modeling problems unlike the complex neural EOT/SB solvers which we mention in §4. But this is very natural: default Gaussian mixture model for density estimation is also not used for modeling complex real-data distributions, e.g., images. At the same time, such models still play irreplaceable role in many smaller scale problems due to its extremal simplicity and ease of use. Therefore, we hope that our solver will play an analogous role in the field of computational continuous EOT/SB.

4. Our work considers SB with the Wiener prior $W^\epsilon$, which is equivalent to EOT with the quadratic cost $c(x, y) = \frac{1}{2}\|x_0 - x_1\|^2$ and entropic regularization value $\epsilon$. In this case, the Gaussian mixture parameterization (9) is useful as it provides the closed-form expression for conditional distributions $\pi_\theta(x_1|x_0)$ and drift $g_\theta$ (Proposition 3.2). This is due to the fact that the product of the unnormalized Gaussian mixture density $v_\theta$ with the unnormalized normal density $\exp(-\frac{1}{2}\|x_0 - x_1\|^2)$ is again a Gaussian mixture. We do not know if our light solver can be easily generalized to more general costs or priors. We leave this question open for future studies.

**Broader Impact**. Deep learning models become more complex, and it may negatively affect the Earth's ecology as the required computational clusters need increasing amounts of energy to work and water to cool them. In fact, sometimes deep neural networks (DNNs) are applied when they may be unnecessary, so they pointlessly burn resources. Our work investigates SB and its applications to moderate-dimensional data, where DNNs are widely used. Our proposed light solver demonstrates that SB can be well-learned without DNNs and in a few minutes even on CPU. This helps to avoid time-consuming GPU-training of previous solvers and decrease the negative impact on nature.

Potential negative broader impact of our research is the same as that of most of the other ML researches. Namely, the constant advances in the field of the AI and the implementation of ML models in production pipelines may lead to transformation or replacement of some jobs in industry.

## H  ADDITIONAL EXPERIMENTAL RESULTS

### H.1  DEPENDENCE ON THE PARAMETER $\epsilon$.

In Figure 5, we show how the solution learned by LightSB depends on the parameter $\epsilon$ in the *Adult→child* experiment. As expected, the diversity increases with the increase of $\epsilon$.

### H.2  ADDITIONAL IMAGE GENERATION RESULTS.

In Figure 6 we show more samples from LightSB trained on every considered image setup.

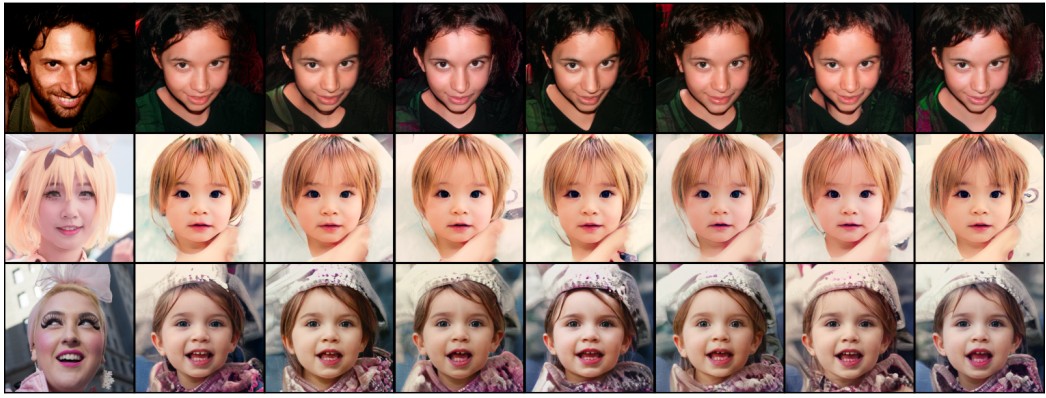

(a) LightSB *Adult → Child*, $\epsilon = 0.1$. Almost no diversity.

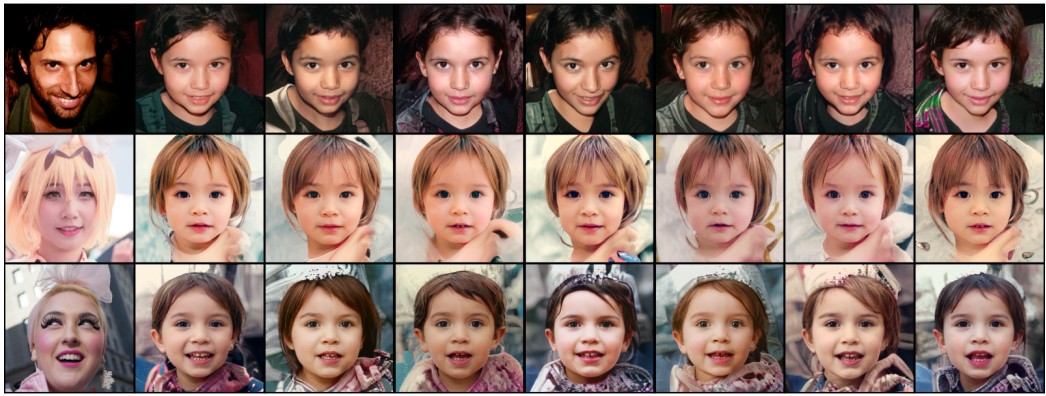

(b) LightSB *Adult → Child*, $\epsilon = 0.5$. Resonable diversity.

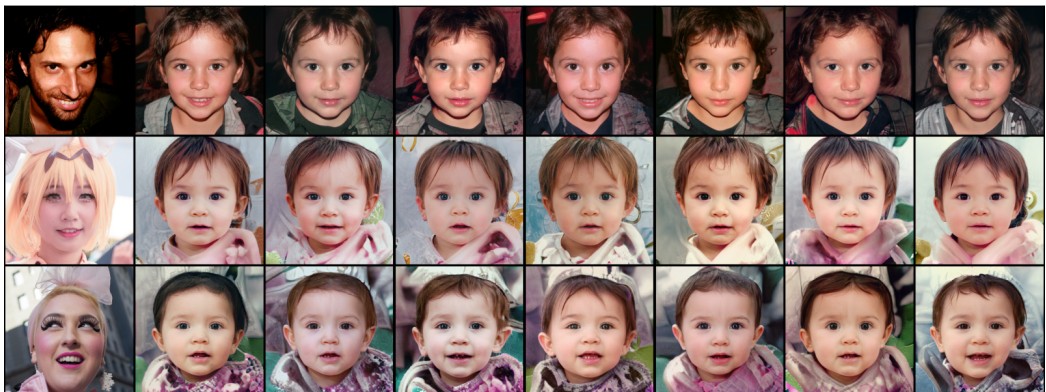

(c) LightSB *Adult → Child*, $\epsilon = 1.0$. Moderate diversity.

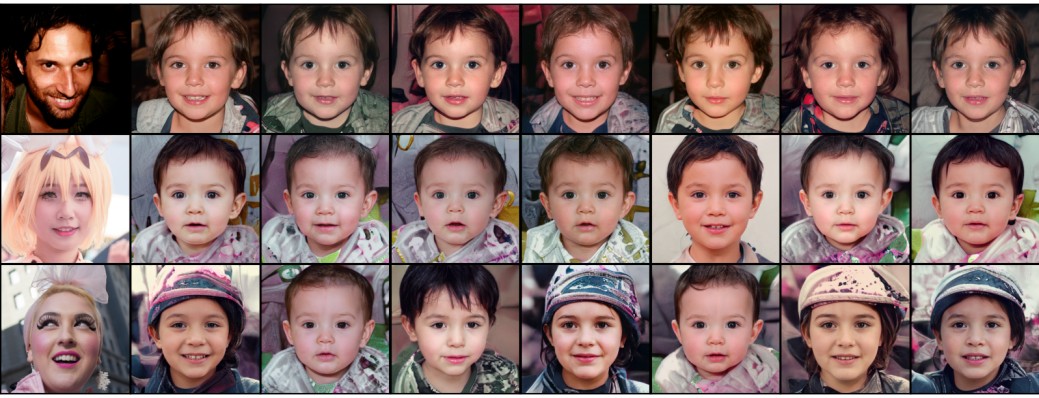

(d) LightSB *Adult → Child*, $\epsilon = 10.0$. High diversity.

Figure 5: LightSB *Adult → Child* for different $\epsilon$.

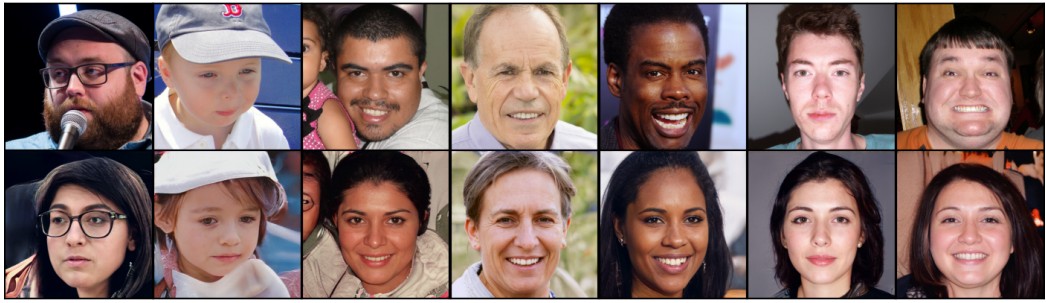

(a) LightSB *Male → Female*, $\epsilon = 0.1$ more samples.

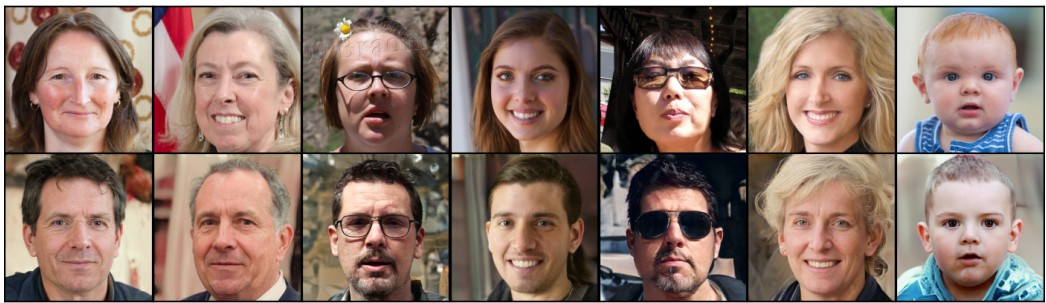

(b) LightSB *Female → Male*, $\epsilon = 0.1$ more samples.

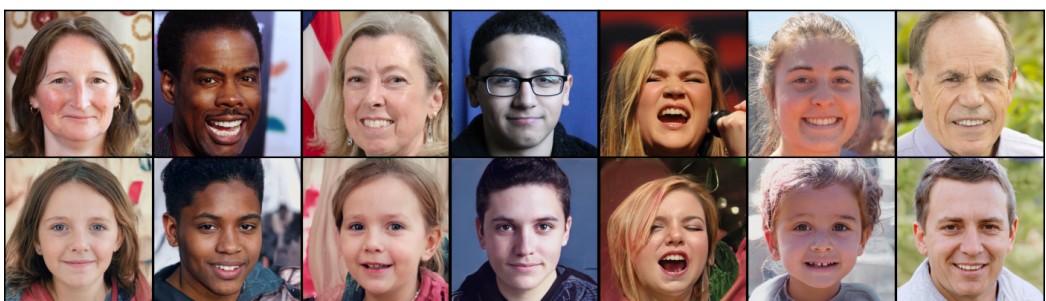

(c) LightSB *Adult → Child*, $\epsilon = 0.1$ more samples.

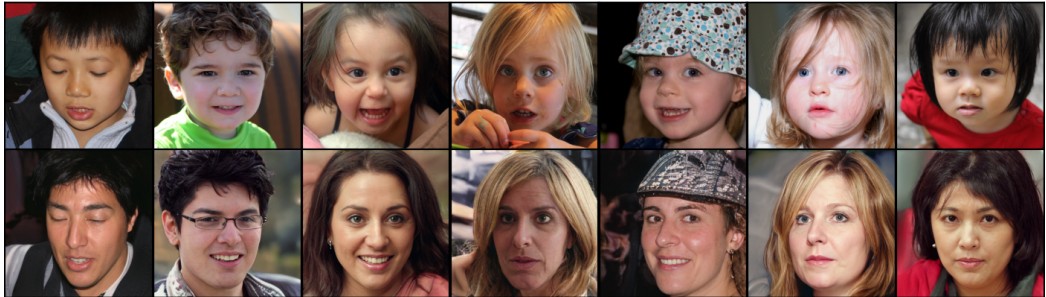

(d) LightSB *Child → Adult* more samples.

Figure 6: LightSB more samples for every considered setup.

