# OpenReview forum: "Light Schrödinger Bridge"
_ICLR.cc/2024/Conference — ICLR 2024 poster_

### Official Review · Reviewer_o21Y · 2023-10-28

**Soundness:** 3 good
**Presentation:** 4 excellent
**Contribution:** 3 good
**Rating:** 8
**Confidence:** 4

**Summary:**

The paper proposes a fast, light-weighted, solver for Schrödinger bridge by using a Gaussian mixture parametrization on the energy potentials. This simplifies computation of the static EOT map that would otherwise be intractable, resulting in efficient learning process. Extensive experiments are conducted on low/mid dimensional benchmark, single-cell dataset, and unpaired image translation in latent space.

**Strengths:**

- The proposed method is notably simple and elegant. It effectively combines key insights from previous works and elevates them to address a significant problem within the SB community.

- The paper is well-written, and the thorough comparison to related works, especially in Table 1, is particularly valuable. Additionally, the comprehensive discussions in the appendix about limitations and broader impacts are appreciated.

**Weaknesses:**

- The proposed method on image dataset requires pretrained latent space (512 dimension) that is already structurally informative.

- On the discussion of tractable / real-world SB given pairing, a few important references such as "Aligned SB" and "image-to-image SB" are missing.

- Given that the proposed method is computationally light weighted, it'll be beneficial to have some quantitative comparison (actual runtime, memory etc) to prior works.

- All "Schrodinger" should be changed to Schrödinger.

**Questions:**

- Can the author provides image experiments without the latent space? I suggest smaller dataset such as AFHQ 32 or 64 for faster evaluation given the limited rebuttal period. Otherwise, can the author provide and include discussions on the scalability of the proposed method? Given Thm 3.4, it seems like the method could be applied to these scenarios.

---

> ### Author Response · Authors · 2023-11-22
>
> Dear Reviewer o21Y, Thank you for your comments. Here are the answers to your questions.
>
> **(1) The proposed method on image dataset requires pretrained latent space (512 dimension) that is already structurally informative.**
>
> We agree with the reviewer.
>
> **(2) Given that the proposed method is computationally light weighted, it'll be beneficial to have some quantitative comparison (actual runtime, memory etc) to prior works.**
>
> Following your question, we considered several new high-dimensional setups, provided the qualitative comparisons with several EOT/SB solvers and compared their training time. Please see **Q1 in the general answer** to all the reviewers.
>
> **(3) Can the author provides image experiments without the latent space? (...). Otherwise, can the author provide and include discussions on the scalability of the proposed method? Given Thm 3.4, it seems like the method could be applied to these scenarios.**
>
> Although theoretically our parameterization could approximate any SB given sufficiently many components in the mixture, practically the required number of such components may bee too large.
>
> Please note that in **Appendix F** (limitations, point 3), we already emphasized that we develop a solver for SB applications in moderate-dimensional spaces, not for large-scale generative modeling. Just like the basic Gaussian mixture model is **not used** for generative modeling, our solver is **not aimed** for learning large-scale generative models. Still the basic Gaussian mixture is a very easy-to-use baseline for many data analysis tasks, so we do believe that our solver will play an analogous role in the field of SB. Specifically, we already see that despite its simplicity it outperforms many methods.
>
>
> **(4) All "Schrodinger" should be changed to Schrödinger.**
>
> Thanks for noting. Done.
>
> **(5) SB given pairing (...) references such as "Aligned SB" and "image-to-image SB" are missing**
>
> Following your suggestion, we added the references to **Section 4** (related works) of the revised paper.
>
> **Concluding remarks**.
> We would be grateful if you could let us know if the explanations we gave have been satisfactory in addressing your concerns about our work. We are also open to discussing any other questions you may have.

---

> > ### Comment · Reviewer_o21Y · 2023-11-23
> > **Thanks for the response**
> >
> > I thank the author for the response. I keep my current score.

---

### Official Review · Reviewer_KWE5 · 2023-10-30

**Soundness:** 3 good
**Presentation:** 3 good
**Contribution:** 3 good
**Rating:** 5
**Confidence:** 3

**Summary:**

This paper proposes a novel method to approximate the Schrödinger Bridge problem. Following the connections of the Schrödinger Bridge problem to entropic optimal transport, the authors want to learn the true entropic OT plan. They parametrize the optimal entropic OT plan as the product of two probabilities, the source measure $p_0$ and a conditional plan $\pi(x_1|x_0)$ (up to a normalization constant). Following this novel parametrization, they show that they can optimize the parametrized plan without the knowledge of the true entropic OT plan. To deal with the normalization constant, they use a mixture of Gaussian representations to get a tractable form of each of the components. They then explain the inference and training procedure. Finally, they perform several experiments to show the practicability of the proposed method (2D synthetic data, an EOT benchmark, single-cell dynamics and unpaired image-to-image experiment).

**Strengths:**

i) The parametrization of the entropic OT plan is novel (to the best of my knowledge) and interesting.

ii) The derived loss is very interesting as learning the parametrized model does not require knowing the true entropic OT map.

iii) The discussion on the normalization constant and the proposed parametrization (with a mixture of Gaussians) to overcome this issue is appealing (even if the computation of the normalization constant is based on existing works).

iv) The method has been tested in different experiments.

v) The paper is clear and easy to read.

**Weaknesses:**

1. In the single-cell experiments, the competitor results are taken from another paper [Tong et al., 2023]. As the evaluation is performed on a leave-one-domain out and training on the others, it is questionable that the training procedure was the same. I believe that some of these competitors should be reproduced by the authors to ensure that the training setting is similar, especially as the unpaired image-to-image experiment is a quantitative experiment without competitors.

2. The experiments were performed in a relatively small dimensional setting: the unpaired image-to-image experiments used a pre-trained feature extractor of dimension 512, the single-cell data experiment used the representation of the 5 whitened principal components (ie dimension of 5), and the EOT benchmark data were (at most) of dimension 128. Therefore, the question of the performance of the proposed method in high-dimensional data is legitimate, especially as the authors use a mixture of Gaussian representation.

3. Little is said about the statistical estimation (with respect to the data dimension and sample size) of the proposed parametrization to the true entropic OT plan $\pi^\star$. It is known that it also suffers from the curse of dimension [2,3].

4. Some related work is missing and should be discussed and mentioned [1,2,3,4]

[1] Stochastic optimization for large-scale optimal transport, Genevay et al, Neurips 2016
[2] On the sample complexity of entropic optimal transport, Rigollet et al.
[3] Minimax estimation of smooth optimal transport maps, Hutter et al.
[4] Learning with minibatch Wasserstein, Fatras et al., Aistats 2020

**Questions:**

1. How does your method depend on the dimensionality of the data? I think that this is a legitimate question, especially with the normalization constant being approximated by a mixture of Gaussians. I recommend adding more dimensionality in some experiments (like single-cell by considering a larger number of principal components. Maybe 5, 50, 200, 500, 1000?) to study how the proposed method performs as the dimension grows. I also recommend adding this discussion to the limitation paragraph in Appendix E.

2. On the single-cell experiment of the unpaired image-to-image experiment, could you show the training speed to reach convergence and the number of iterations it took? I checked the appendix and I did not find such plots. Maybe it would be interesting to compare to other OT solvers (sinkhorn or stochastic variants) on a simple problem to see the different behaviour. (I acknowledge that the Sinkhorn solvers are only usable in a discrete setting.)

3. I found the limitation discussion in Appendix E interesting and it could have been in the main paper. I recommend moving it to the main paper. Maybe some discussions about related work could go in the appendix instead.

I am ready to reconsider my score if the authors can reproduce some of the competitor results and consider the highest dimensional setting of single-cell data experiments (dimension of +1000). It will depend on how well the proposed algorithm will behave on medium dimensionality.


----- EDIT POST REBUTTAL -----

Thank you for your answer. I have read the rebuttal.

[Single-cell experiments] Thank you for the novel experiments on the single-cell trajectory problem. I find them interesting and encouraging, especially with the different dimensions.

I understand the motivation for using a different metric than W_1 to compare the different approaches. However as it is not the standard metric, it is hard for me to compare your method with other standard single-cell trajectory methods. As the authors-reviewer discussion has ended (due to a late rebuttal submission), I would have appreciated seeing the Wasserstein 1 metric for the different approaches. Indeed, the novel dataset was first considered by [Tong et al., 2023] where they used the W_1 distance. As the authors did not reproduce their method with the considered metric, it is hard to compare with more standard approaches on single-cell datasets like ODE/SDE-based approaches [Tong et al., 2023]. Unfortunately, two of the considered competitive methods are not standard in the single-cell trajectory literature. Therefore, I still think that the experimental section lacks reproduced competitors especially as the metrics are different.

[Novel theoretical section] Thank you for the novel appendix H. It brings some light to the proposed methodology. I suggest the authors include Theorem H.1 in the main paper.

In my opinion, this is a borderline paper. I acknowledge that the proposed approach is new and interesting but I still think that it lacks a more rigorous experimental section to understand its practical performance with the current literature. Therefore, I will keep my score.

---

> ### Author Response · Authors · 2023-11-22
>
> Dear Reviewer KWE5, Thank you for your comments. Here are the answers to your questions.
>
> **(1) the single-cell experiments (...) relatively small dimensional setting (...) I am ready to reconsider my score if the authors can reproduce some of the competitor results and consider the highest dimensional setting of single-cell data experiments**
>
> Following your question, we considered several new high-dimensional setups, provided the qualitative comparisons with several EOT/SB solvers and compared their training time. Please see **Q1 in the general answer** to all the reviewers.
>
> **(2) Little is said about the statistical estimation (with respect to the data dimension and sample size) of the proposed parametrization to the true entropic OT plan (...)**
>
> Following your question, we added a new **Appendix H** where we demonstrated that our solver experiences a usual **parametric** convergence rate in the statistical error  ($\sim \frac{1}{\sqrt{N}}+\frac{1}{\sqrt{M}}$, where $N,M$ are sample sizes), please see **Q2 in the general response** to all the reviewers. In the future, it would be interesting to further study the trade-off between the statistical and approximation errors, their dependence on $K$ (number of components) and $D$ (dimension).
>
> **(3) Some related work is missing and should be discussed and mentioned [1,2,3,4]**
>
> Please note that work [3] has already been mentioned in **Appendix E** (related work: discrete OT). Following your suggestion, we additionally mentioned [2,4] in **Appendix E** and [1] in **Section 4** (related work: continuous EOT).
>
> **(4) Comparison with Sinkhorn**
>
> We have done comparison with the (discrete) Sinkhorn algorithm as a part of our additional comparison, please see again **Q2 in the general answer** to all the reviewers and Appendix C.4. We see that for small $\epsilon$ our solver yield even faster convergence than the Sinkhorn algorithm.
>
> **Concluding remarks**.
> We would be grateful if you could let us know if the explanations we gave have been satisfactory in addressing your concerns about our work. If so, we kindly ask that you consider increasing your rating. We are also open to discussing any other questions you may have.
>
> **References**
>
> [1] Stochastic optimization for large-scale optimal transport, Genevay et al, Neurips 2016
>
> [2] On the sample complexity of entropic optimal transport, Rigollet et al.
>
> [3] Minimax estimation of smooth optimal transport maps, Hutter et al.
>
> [4] Learning with minibatch Wasserstein, Fatras et al., Aistats 2020

---

> > ### Author Response · Authors · 2023-11-23
> >
> > Dear reviewer KWE5, we would like to add some clarifications regarding new experiments that we discussed in our previous response to you.
> >
> > For the new experiments with single cell data (Section 5.3) with dimensions: 50, 100, 1000, we consider the energy distance metric instead of Wasserstein-1 used in the previous single cell setup (Appendix B). Our motivation lies in the statistical properties of estimating these metrics by samples. The Wasserstein-1 metric has a bad sample complexity $O(\frac{1}{n^{\frac{1}{D}}})$, while the energy distance (a variant of the maximum mean discrepancy metric) has a sample complexity $O(\frac{1}{\sqrt{n}})$ and allows unbiased estimates from samples [1].
> >
> > [1] Genevay A. et al. Sample complexity of sinkhorn divergences //The 22nd international conference on artificial intelligence and statistics. - PMLR, 2019. - С. 1574-1583.

---

### Official Review · Reviewer_cSx4 · 2023-10-31

**Soundness:** 1 poor
**Presentation:** 3 good
**Contribution:** 1 poor
**Rating:** 5
**Confidence:** 4

**Summary:**

This paper proposes a fast Schrodinger Bridge (SB) solver based on the parameterization of Schrodinger potentials. SB is a dynamic version of the Entropic Optimal Transport (EOT), and there exists plenty of solvers for these problems, they requires complex parameterization by the form of neural networks / or is sensitive with the entropy regularization parameters, and thus are costly in terms of evaluation on larger scale datasets. The authors of this work propose a solution to this problem by consider a settings where the continuous SB is associated with a Wiener prior (as a reference measure). The key idea is to consider the parameterization of the Schrodinger potential as a mixture of Gaussian, and rewrite the SB optimization objective following this, with easy to compute mean and (scalar diagonal) covariance matrix. The authors provide some theoretical analysis of their algorithm, along with empirical demonstrations on synthetic dataset and realistic dataset (single cell data population dynamic and image-to-image translation).

**Strengths:**

- Algorithms based on well-studied theory of SB and EOT.
- The paper is well-written with clear structure.
- Well-motivated problem: lightweight solver is much needed for problem that usually requires heavy computational power.

**Weaknesses:**

* **Novelty of the paper:** it seems like the paper is just a combination of the two previous works [1, 2]. Admittedly, the authors have clearly elaborated in their paper of such cases, but I do not see the clear novelty in the methodology part. Even on the proof of the universal theorem of the Gaussian mixture model, one can see that half of it is straightforward calculation from the two aforementioned paper.
* **Unclear benefit the of LightSB solver in realistic setting:** for the single cell dataset using W1 distance as a metric, LightSB's performance leaves some gaps with the two best methods, however I do not see author's comment about this. The other benchmark on unpaired image-to-image translation is very hard to judge, as for this task there is no quantitative metric to compare with other solvers. I do not know why the authors omit unconditional image generation tasks, as this is an important and popular benchmark that has FID as a standard metric. Morevoer, there have already exist results on some of the neural EOT solvers or the diffusion SB solver (using iterative proportional fitting)/diffusion SB matching (using iterative Markovian fitting), or flow matching/rectified flow in this task, so it is easy to compared with the baseline.
*  **Questionable theoretical result:** in the proof the universality of the Gaussian mixture parameterization for SB (theorem 3.4), in the paragraph below equation (29), I fail to understand why the authors wrote

> "Besides, it also has scalar covariances of its components because multiplier $exp(− |x_1|^2 /2\epsilon )$’s covariance is scalar itself"

, but what I understood in the paper's settings is that $x_1 \sim \pi_1$ is an unknown distribution, without assumptions on its parametric form. This is a key argument, as without it the factorization failed to be what the authors claimed in the statement of the theorem, I hope the authors could clarify this to me, otherwise it would be a hole in the proof of an important analysis of the paper.

[1] Petr Mokrov, Alexander Korotin, and Evgeny Burnaev. Energy-guided entropic neural optimal transport. arXiv preprint arXiv:2304.06094, 2023.
[2] Nikita Gushchin, Alexander Kolesov, Alexander Korotin, Dmitry Vetrov, and Evgeny Burnaev. Entropic neural optimal transport via diffusion processes. In Advances in Neural Information Processing Systems, 2023a.

**Questions:**

See weaknesses section.

---

> ### Author Response · Authors · 2023-11-22
> **Part I**
>
> Dear Reviewer cSx4, Thank you for your comments. Here are the answers to your questions.
>
> **(1) Novelty [...]: [...] the paper is just a combination of the two previous works [1, 2]. Admittedly, the authors have clearly elaborated in their paper of such cases, but I do not see the clear novelty in the  methodology part.**
>
> In the mentioned paper [2], the authors propose a scheme that allows to construct, for a given distribution $p_0$, a new distribution $p_1$ with a known conditional entropy regularized optimal transport (entropic OT) plan $\pi^*(y|x)$. However, the authors **did not propose any algorithm** to solve entropic OT between two given distributions $p_0$ and $p_1$.
>
> In the other paper [1], the authors propose a new **algorithm** based on the connection between entropic OT in dual form and energy based models. Due to the use of energy-based like objective, they need to use Langevin sampling at both training and inference steps, **which is very time consuming.**
>
> In our paper, we propose **an lightweighted and fast algorithm that allows to directly minimize the KL-divergence KL** $(\pi^*||\pi\_{\theta})$ between the parametric approximation $\pi\_{\theta}$ and the solution to the entropic OT $\pi^*$ without knowing $\pi^*$ and even without needing to have samples of it (Proposition 3.1). Furthermore, we show that by using Gaussian mixture parameterization, this KL minimization can be performed by a standard gradient descent algorithm without the need to use time-consuming Langevin dynamics. Moreover, we proved that such parameterization has a universal approximation property (Theorem 3.4) similar to the Gaussian mixture model.
>
> We believe this is a sufficient amount of novelty to demonstrate that our work is not just a combination of prior works.
>
> **(2) Even on the proof of the universal theorem of the Gaussian mixture model, one can see that half of it is straightforward calculation from the two aforementioned paper.**
>
> We respectfully disagree. The only thing where the proof may intersect with the above-mentioned papers [1,2] is the usage of the duality form from weak OT. This generic known result just serves an intermediate step to guarantee the existence of the nearly-optimal continuous dual variable (stage 1 of the proof). The main (*the trickiest and the longest*) part of the proof (stage 2) is to demonstrate that the dual form can be approximated arbitrarily well with the Gaussian mixtures. This is highly **non trivial** due to the necessity to simultaneously control the approximation error for the potential and its $C$-transform. To our knowledge, this principal analysis is completely novel.
>
> *We emphasize that, to our knowledge, our Theorem 3.4 is the first ever universal approximation result for the SB.*
>
> **(3) Unclear benefit the of LightSB solver in realistic setting: for the single cell dataset using W1 distance as a metric, LightSB's performance leaves some gaps with the two best methods, however I do not see author's comment about this.**
>
> The main benefit of our solver is the simplicty and its speed. First, it is as simple as the well-celebrated Gaussian mxiture for the density estimation. Second, our solver converges on CPU in 1 minute while the rest methods require longer training times. As one may see, e.g., from Tables 3 and 4, despite its simplicity, *our method already beats many existing complex approaches.* Please also see **Q1 in the general response** to all the reviewers.
>
> **(4) The other benchmark on unpaired image-to-image translation is very hard to judge (...). I do not know why the authors omit unconditional image generation tasks (...).**
>
> In our paper, we emphasize that we develop a solver for SB applications in moderate-dimensional spaces, not for large-scale generative modeling. Just like the basic Gaussian mixture model is **not used** for generative modeling, our solver is **not aimed** for learning large-scale generative models. Still the basic Gaussian mixture is a very easy-to-use baseline for many data analysis tasks, so we do believe that our solver will play an analogous role in the field of SB. Specifically, we already see that despite its simplicity it outperforms many methods.

---

> > ### Author Response · Authors · 2023-11-22
> > **Part II**
> >
> > **(5) Questionable (...) universality of the Gaussian mixture parameterization (...) I fail to understand why the authors wrote "Besides, it also has scalar covariances of its components because multiplier $exp(-|x_1|^2/2\epsilon)$'s’s covariance's scalar itself", but what I understood in the paper's settings is that  is an unknown distribution, without assumptions on its parametric form. This is a key argument, as without it the factorization failed to be what the authors claimed in the statement of the theorem, (...).**
> >
> > We think there is a misunderstanding of our Theorem 3.4 from your side and are happy to give clarifications below.
> >
> > **We do not make any assumptions on the form of distributions $p_0,p_1$ and the OT plan $\pi^{*}$**. We only require $p_0,p_1$ to be supported on the compact sets. What we do is we prove the existence of a parametric approximator $v_{\theta}$ for the adjusted Schrodinger potential which yields the approximation $\pi_{\theta}\approx \pi^{*}$ of the OT plan. This **approximator has a specific parametric form**, namely, $v_{\theta}$ is an unnormalized Gaussian mixture (GM) with scalar covariances of components.
> >
> > In the proof of our universal approximation theorem, we first pick a GM approximator $v_{\widetilde{\theta}}$ (eq. 28) with scalar covariance for the function $\exp(\frac{\widetilde{f}(x_1)}{\epsilon})$ which is introduced during the proof. Then we define a new function by $v_{\theta}(x_1)=v_{\widetilde{\theta}}(x_1)\cdot \exp(-\frac{\|x_1\|^{2}}{2\epsilon})$. Hence, the statement *"it ($v_{\theta}$) also has scalar covariances of its components because multiplier $exp(-|x_1|^2/2\epsilon)$'s covariance's scalar itself*" is related exclusively to the parametric approximator $v_{\theta}$, not to $p_0,p_1$ or $\pi^{*}$.
> >
> > *As per your request, in the revised paper, we added the detailed derivation of this statement to the proof of this our Theorem 3.4 (Appendix A).*
> >
> > **Concluding remarks**.
> > We would be grateful if you could let us know if the explanations we gave have been satisfactory in addressing your concerns about our work. If so, we kindly ask that you consider increasing your rating. We are also open to discussing any other questions you may have.
> >
> > **Additional references.**
> >
> > [1] Petr Mokrov, Alexander Korotin, and Evgeny Burnaev. Energy-guided entropic neural optimal transport. arXiv preprint arXiv:2304.06094, 2023.
> >
> > [2] Nikita Gushchin, Alexander Kolesov, Alexander Korotin, Dmitry Vetrov, and Evgeny Burnaev. Entropic neural optimal transport via diffusion processes. In Advances in Neural Information Processing Systems, 2023a.

---

> > > ### Comment · Reviewer_cSx4 · 2023-11-23
> > > **Thank you for your rebuttal**
> > >
> > > My concern regarding the proof of the universality is cleared. I am therefore raise the score to 5, as I see the paper's merit and clear motivation for solving the SB problem fast and requires less resources than other baselines, but still have concern over the novelty of this paper and the lack of unconditional image generation task.
> > >
> > > I have a quick quetion regarding the new benchmarks in Section 5.3 however: why did the authors use different metric (energy distance) than before, which is a variant of Bures-Wasserstein metric, that in my opinion is fairer to compare (assumed) Gaussian distributions? For three different benchmarks now I see the authors used three different distances as metrics.
> > >
> > > Note that since the author submitted the rebuttal very late into the discussion phase, I have not time to review the additional theoretical section in the Appendix yet (regarding statistical property).

---

> > > > ### Author Response · Authors · 2023-11-23
> > > >
> > > > We are pleased that our explanations help to address concerns about our work.
> > > >
> > > > We choose different metrics based on the different nature of the data considered.
> > > >
> > > > - For the benchmark setup (Section 5.2), we use the same metric proposed by the authors of the benchmark [1], since they have done extensive comparisons for many other EOT/SB solvers.
> > > >
> > > > - For the previous 5-dim single-cell setup (Appendix B), we used the Wasserstein-1 metric, since it has been used in this setup in many previous papers and works reasonably well for low-dimensional data.
> > > >
> > > > - For the new single-cell setup (Section 5.3) with dimensions: 50, 100, 1000, we choose the energy distance instead of the Wasserstein-1 metric, since the Wasserstein-1 metric has a terrifically bad sample complexity $O(\frac{1}{n^{\frac{1}{D}}})$, while the energy distance (a variant of the maximum mean discrepancy metric) has a sample complexity $O(\frac{1}{\sqrt{n}})$ and admits unbiased estimates from samples [1].
> > > >
> > > > [1] Gushchin N. et al. Building the Bridge of Schrödinger: A Continuous Entropic Optimal Transport Benchmark // In Advances in Neural Information Processing Systems, 2023.
> > > >
> > > > [2] Genevay A. et al. Sample complexity of sinkhorn divergences //The 22nd international conference on artificial intelligence and statistics. - PMLR, 2019. - С. 1574-1583.

---

### Official Review · Reviewer_o9rU · 2023-11-01

**Soundness:** 3 good
**Presentation:** 3 good
**Contribution:** 3 good
**Rating:** 8
**Confidence:** 1

**Summary:**

Current SB solvers often have complex neural parameterization and time-consuming training procedures due to minimax. This work proposed a novel parameterization technique and a non-minimax training method to bypass the issues above, with the aim at offering a lightweight and easy-to-use SB baseline.

**Strengths:**

Strong motivation: the complex neural parameterization and time-consuming training procedures do hinder the application of current SB methods.

Clear presentation: the background knowledge and preliminaries are clearly presented. The connections with important references, such as (Tong et al., 2023), are highlighted and summarized in Table 1. The narrative is highly structural, with words and paragraphs to offer overviews of important sections.

Soundness: the work seems highly self-contained: learning objectives, training/inference strategies, and theoretical properties. Limitations are also well discussed in appendix, some of which are examined through experiments.

**Weaknesses:**

1. Equation (5) seems incomplete or not well-defined. The equation is a crucial part of the methodology, and its clarity is essential for both understanding the method and replicating the results.

2. It is pointed regarding other SB solvers that `they expectedly require time-consuming training/inference procedures.`. The authors state that their method avoids `time-consuming max-min optimization, simulation of the full process trajectories, iterative learning, and MCMC techniques,` which are commonly employed in existing solvers. While this claim holds conceptual interest, empirical evidence to substantiate this would significantly strengthen the paper.

In summary, although there exist some issues with this work, most of them seem correctable. I look forward to seeing the author feedback and the revised manuscript.

**Questions:**

See the weaknesses above.

---

> ### Author Response · Authors · 2023-11-22
> **Rebuttal answer**
>
> Dear Reviewer o9rU, thank you for your comments. Here are the answers to your questions.
>
> **(1) Equation (5) seems incomplete or not well-defined.**
>
> It seems like we do not completely understand your comment. Equation (5) just explains that we would like to find the best approximation (in terms of KL divergence) of optimal plan $\pi^{*}$ in a given set of distributions $\pi_{\theta}$ indexed by a parameter $\theta$. The particular parameterization is introduced in the following narrative (equations 7 and 9).
>
> **(2) It is pointed regarding other SB solvers that they expectedly require time-consuming training/inference procedures. (...) While this claim holds conceptual interest, empirical evidence to substantiate this would significantly strengthen the paper.**
>
> Following your question, we considered several new high-dimensional setups, provided the qualitative comparisons with several EOT/SB solvers and compared their training time. Please see **Q1 in the general answer** to all the reviewers.
>
> **Concluding remarks**.
> We would be grateful if you could let us know if the explanations we gave have been satisfactory in addressing your concerns about our work. We are also open to discussing any other questions you may have.

---

> > ### Comment · Reviewer_o9rU · 2023-11-23
> >
> > Hi authors,
> >
> > Thanks for the detailed explaination and I have read all of them. I really appreciate you taking the time to answer my question. I will keep my rating and I do think this is an interesting and insightful paper.

---

### Official Review · Reviewer_rvYk · 2023-11-04

**Soundness:** 3 good
**Presentation:** 3 good
**Contribution:** 3 good
**Rating:** 8
**Confidence:** 3

**Summary:**

Summary
1) The authors propose a novel light solver for continuous SB with the Wiener prior, i.e., EOT with the quadratic transport cost. The solver has a straightforward non-minimax learning objective and uses the Gaussian mixture parameterization for the EOT/SB, which avoids the time-consuming max-min optimization, simulation of the full process trajectories, iterative learning, and MCMC techniques that are in use in existing continuous solvers.
2) The authors show that their novel light solver provably satisfies the universal approximation property for EOT/SB between the distributions supported on compact sets.
3) The authors demonstrate the performance of the light solver in a series of synthetic and real-data experiments, including the ones with the real biological data considered in related works.

**Strengths:**

Strengths:
- The authors provide a light solver for SB that does not rely on neural network parametrization.
- This work provides a universal approximation for the solver which seems to be rather non-trivial. But to be honest, I'm not familiar with the proof so I'm not sure of the technical depth.
- A minor thing but I really appreciate the authors giving a very clear description of the limitations of the solver.

Following the rebuttal, the authors also provided statistical guarantees for the method.

**Weaknesses:**

Weaknesses:
This work does not have a finite-time nor finite-sample convergence guarantee. I believe that with additional assumptions one can obtain convergence guarantees as in [1]. Furthermore, there have been multiple works on the sampling complexity of quadratic cost EOT, and given the equivalence between EOT and SB, I believe that obtaining guarantees should be feasible.

I also think that the experiments are not comprehensive enough. While the results are nice, I'm wondering what is the runtime performance of the light solver against other solvers. I cannot vouch for the method if I don't know how the method would improve in terms of the quality of the result and the computational cost.

The authors did mention that the work relies on Gaussian mixture parameterization and the entropic cost. It is indeed a limitation but personally, I think it is fine. Nevertheless, I will still raise this as a potential weakness.

All and all, I like the paper but it does not convince me enough to vouch for its acceptance yet.

Minor comments:
I think the literature on EOT is not sufficient. The authors should include gradient-based EOT solvers such as [2] (both of these methods achieved the optimal O(n^2/eps) complexity, which is stronger than that of APDAGD and seems to have good performance) and a gradient-based entropic UOT solver [3].

References:
[1] Chen, Y., Deng, W., Fang, S., Li, F., Yang, N. T., Zhang, Y., … Nevmyvaka, Y. (2023). Provably Convergent Schrödinger Bridge with Applications to Probabilistic Time Series Imputation. Retrieved from http://arxiv.org/abs/2305.07247
[2] Xie, Y., Luo, Y., & Huo, X. (2023). An Accelerated Stochastic Algorithm for Solving the Optimal Transport Problem. Retrieved from http://arxiv.org/abs/2203.00813
[3] "On Unbalanced Optimal Transport: Gradient Methods, Sparsity and Approximation Error".
Quang Minh Nguyen, Hoang Huy Nguyen, Lam Minh Nguyen, Yi Zhou

**Questions:**

Questions:
How is this method compared to Langevin dynamics/Langevin Monte Carlo methods in terms of runtime and empirical results?

---

> ### Author Response · Authors · 2023-11-22
> **Rebuttal answer**
>
> Dear Reviewer rvYk, thank you for your comments. Here are the answers to your questions and comments.
>
> **(1) This work does not have a finite-time nor finite-sample convergence guarantee. I believe that with additional assumptions one can obtain convergence guarantees as in [1].**
>
> Thanks for the comment, we added the suggested paper to Section 4 (related works). To be honest, we do not think that the results of paper [1] are of close relevance to our paper. The mentioned paper estimates the errors of the approximate Iterative Proportional Fitting (IPF) procedure. One of the key advantages of **our light solver** is it is free from IPF-like procedures, it *directly optimizes KL* w.r.t. the optimal plan in the certain class of parameterized plans *without any IPF-like iterations*. That is, it seems like there is no need to analyze convergence in the sence of [1].
>
> **(2) ... Furthermore, there have been multiple works on the sampling complexity of quadratic cost EOT, and given the equivalence between EOT and SB, I believe that obtaining guarantees should be feasible.**
>
> Following your question, we added a new **Appendix H** where we demonstrated that our solver experiences a usual **parametric** convergence rate in the statistical error  ($\sim \frac{1}{\sqrt{N}}+\frac{1}{\sqrt{M}}$, where $N,M$ are sample sizes), please see **Q2 in the general response** to all the reviewers. In the future, it would be interesting to further study the trade-off between the statistical and approximation errors, their dependence on $K$ (number of components) and $D$ (dimension).
>
> **(3) (...) I'm wondering what is the runtime performance of the light solver against other solvers. (...).**
>
> Following your question, we considered several new high-dimensional setups, provided the qualitative comparisons with several EOT/SB solvers and compared their training time. Please see **Q1 in the general answer** to all the reviewers.
>
> **(4) The authors should include gradient-based EOT solvers [2, 3].**
>
> Thanks for noting, we added [2,3] to the discussion of OT solvers in **Appendix D**. Please note that these solvers are discrete solvers and they are of limited relevance to our continuous setup (Section 2).
>
> **Concluding remarks**.
> We would be grateful if you could let us know if the explanations we gave have been satisfactory in addressing your concerns about our work. If so, we kindly ask that you consider increasing your rating. We are also open to discussing any other questions you may have.
>
> **References.**
>
> [1] Chen, Y., Deng, W., Fang, S., Li, F., Yang, N. T., Zhang, Y., … Nevmyvaka, Y. (2023). Provably Convergent Schrödinger Bridge with Applications to Probabilistic Time Series Imputation. Retrieved from http://arxiv.org/abs/2305.07247
>
> [2] Xie, Y., Luo, Y., Huo, X. (2023). An Accelerated Stochastic Algorithm for Solving the Optimal Transport Problem. Retrieved from http://arxiv.org/abs/2203.00813
>
> [3] "On Unbalanced Optimal Transport: Gradient Methods, Sparsity and Approximation Error". Quang Minh Nguyen, Hoang Huy Nguyen, Lam Minh Nguyen, Yi Zhou
>
> [4] Mokrov et. al. (2023). Energy-guided Entropic Neural Optimal Transport. arXiv preprint arXiv:2304.06094.

---

> > ### Comment · Reviewer_rvYk · 2023-11-23
> > **Thanks for the response**
> >
> > Dear authors,
> >
> > Thank you for your detailed response. I think the results are very nice, from both a theoretical and practical standpoint. Although the added experimental result in Appendix B is not too impressive, I understand that it is due to a shortage of time and it does not undercut the main selling point of speed. That being said, I hope that in later iterations of the paper, the authors will add more experimental results and discuss more on the run time of the method.
> >
> > On the other hand, I like the theoretical results of this paper and the authors did provide additional theoretical guarantees as requested and addressed all of my concerns. Thus, I will increase my score. However, I must note that the authors should add and improve the experimental results should this paper get accepted.
> >
> > Best wishes,

---

### Author Response · Authors · 2023-11-22
**General response**

Dear reviewers, thank you for taking the time to review our paper. We are pleased that all the reviewers clearly listed the strengths of our proposed solver which includes the simplicity and elegance of the proposed solver, clarity of the exposition and theoretical justification from the approximation point of view. We really hope that the conceptual similarity of  our light solver to the well-celebrated Gaussian mixture could help our solver to became the standard easy-to-use baseline for SB methods. Please, find the answers to your shared questions below.

**(1) Evaluation on higher dimensions, runtime comparison, etc. (Reviewers rvYk, o9rU, KWE5, o21Y)**

As per request of the reviewers, we evaluate our solver in higher-dimensional setups (DIM=$50, 100, 1000$). We put these additional experiments to the main text in **Section 5.3** (and **Appendix C.4**). The previously considered 5-dimensional setup is moved to **Appendix B**. In this new experiment, we follow the setup from [1] with the data from the Kaggle competition "Open Problems - Multimodal Single-Cell Integration". We solve SB between two time steps $t_0$ and $t_{2}$ and then predict the distribution at the intermediate step $t_1$, see *the details in the revised paper*. As a metric, we use the well celebrated energy distance [2], which admits unbiased estimation from samples, and as baselines, we use several EOT/SB solvers built on different principles (adversarial, Langevin-based, IPF-based), plus we also adapt the discrete Sinkhorn solver for comparison.

*As we report in newly added Table 3, overall our solver is among the best performing; it uses only CPU, while the rest solvers require more time on GPU. In the revised submission, we also added a new reproducible $*.ipynb$ with the code to test our solver on this setup.*

**(2) Statistical properties of the proposed light solver (Reviewers rvYk, KWE5)**

Just like many other works developing parametric (neural) EOT/SB solvers, we primarily focused on developing the computational procedure rather than analysing its statistical properties. Nevertheless, as per the request of the reviewers, **we added new Appendix H with the derivations of the finite-sample convergence properties.**

In short, for our solver it is straightforward to decompose the generalization error into statistical and approximation errors analogously to Theorem 4 in [3]. We already know from **our** Theorem 3.4 that the approximation error can be made arbitrarily small. Furthermore, in our case  (unlike the mentioned paper), the statistical error admits a clear bound. Namely, we establish the usual **parametric** $\sim(\frac{1}{\sqrt{N}}+\frac{1}{\sqrt{M}})$ rate of convergence of the statistical error ($N,M$ are the sample sizes of $X^{0}\sim p_0,X^{1}\sim p_1$).

*This fact distinguishes our solver from prior development in the field of continuous (neural) SB/EOT; they are usually tested only empirically; typically they do not have any established learning guarantees for recovering the EOT plan/SB.*

**Revised paper.** To sum up, the main edits to the paper are highlighted with the **blue color** (newly added) and **orange color** (previous content moved to Appendix):

- New **Section 5.3** + **Appendix C.4** with high-dimensional experiments (Reviewers rvYk, o9rU, KWE5, o21Y).

- New **Appendix H** deriving the generalization properties of our solver (Reviewers rvYk, KWE5).

- Moved initial Section 5.3 to **Appendix B**.

- Added additional clarifications to the proof of Theorem 3.4 in **Appendix A** (cSx4);

- Added various suggested citations (rvYk, KWE5, o21Y).

**References**

[1] Tong A. et al. Simulation-free Schrödinger bridges via score and flow matching //arXiv preprint arXiv:2307.03672. – 2023.

[2] Rizzo, M. L., \& Székely, G. J. (2016). Energy distance. wiley interdisciplinary reviews: Computational statistics, 8(1), 27-38.

[3] Mokrov et. al. (2023). Energy-guided Entropic Neural Optimal Transport. arXiv preprint arXiv:2304.06094.

---

### Meta-Review · Area_Chair_yVVZ · 2023-12-08

**Metareview:**

The authors propose a fast algorithmic approach for Schrodinger bridge by leveraging Gaussian mixture parameterization for the energy potentials. The authors illustrate the advantages of its fast learning process while still provides good solution for Schrodinger bridge in several experiments, including single-cell, unpaired image-to-image translation. Overall, the Reviewers appreciate the novelty of the proposed approach. Additionally, the proposed method is elegant and fast in learning process or Schrodinger bridge. However, the Reviewers also raised concerns about experiments, i.e., baselines, evaluated metrics. Overall, I think this submission is on the borderline, but it seems that the advantages outweigh the raised issues. It is important to propose a fast efficient algorithmic approach in a simple and elegant form for Schrodinger bridge.

I urge the authors to incorporate the Reviewers' comment, and rebuttal discussion in the updated version.

**Justification For Why Not Higher Score:**

+ The proposed ideas are interesting and elegant. I think it is important to propose a fast, efficient algorithm for learning process on Schrodinger bridge problem. However, some issues on the experiments are raised by the Reviewers, e.g., more competitive baselines; evaluation metrics.

**Justification For Why Not Lower Score:**

+ The proposed ideas are interesting and elegant. I think it is important to propose a fast, efficient algorithm for learning process on Schrodinger bridge problem.

---

### Decision · Program_Chairs · 2024-01-16

Accept (poster)